EMBO
Molecular Medicine

# A *Klebsiella pneumoniae* antibiotic resistance mechanism that subdues host defences and promotes virulence

Timothy J Kidd[1,2,3] [ID], Grant Mills[1], Joana Sá-Pessoa[1], Amy Dumigan[1], Christian G Frank[1], José L Insua[1], Rebecca Ingram[1], Laura Hobley[1] & José A Bengoechea[1,*] [ID]

## Abstract

*Klebsiella pneumoniae* is an important cause of multidrug-resistant infections worldwide. Recent studies highlight the emergence of multidrug-resistant *K. pneumoniae* strains which show resistance to colistin, a last-line antibiotic, arising from mutational inactivation of the *mgrB* regulatory gene. However, the precise molecular resistance mechanisms of *mgrB*-associated colistin resistance and its impact on virulence remain unclear. Here, we constructed an *mgrB* gene *K. pneumoniae* mutant and performed characterisation of its lipid A structure, polymyxin and antimicrobial peptide resistance, virulence and inflammatory responses upon infection. Our data reveal that *mgrB* mutation induces PhoPQ-governed lipid A remodelling which confers not only resistance to polymyxins, but also enhances *K. pneumoniae* virulence by decreasing antimicrobial peptide susceptibility and attenuating early host defence response activation. Overall, our findings have important implications for patient management and antimicrobial stewardship, while also stressing antibiotic resistance development is not inexorably linked with subdued bacterial fitness and virulence.

**Keywords** antimicrobial peptides; *Klebsiella pneumoniae*; *mgrB*; polymyxins; virulence
**Subject Categories** Microbiology, Virology & Host Pathogen Interaction

## Introduction

The widespread emergence of multidrug-resistant (MDR) bacterial pathogens is an important public health challenge worldwide (World Health Organization, 2014). Infections with MDR organisms are associated with increased mortality, longer hospital stays and inflated healthcare costs (Lambert *et al*, 2011; Neidell *et al*, 2012; Martin-Loeches *et al*, 2015). Recent data also indicate a trend towards increased antibiotic resistance among cases of community-onset infections (Lim *et al*, 2014; World Health Organization, 2014; Stefaniuk *et al*, 2016). For many bacterial pathogens, particularly Gram-negative organisms, high rates of antimicrobial resistance present limited therapeutic options for treating serious infections.

*Klebsiella pneumoniae* is one of these MDR organisms identified as an urgent threat to human health by the World Health Organization, the US Centers for Disease Control and Prevention and the UK Department of Health. *K. pneumoniae* infections are particularly a problem among neonates, elderly and immunocompromised individuals within the healthcare setting, but this organism is also responsible for a significant number of community-acquired infections including pneumonia and sepsis (Paczosa & Mecsas, 2016; Quan *et al*, 2016). The lineage defined as sequence type (ST) 258 is a notorious example of MDR *K. pneumoniae*; ST-258 frequently carries the *K. pneumoniae* carbapenemase (KPC) gene, as well as numerous other acquired AMR determinants, and has been responsible for outbreaks on several continents (Paczosa & Mecsas, 2016). Colistin is now often considered as the last treatment option for KPC-producing *K. pneumoniae*, but reports of colistin-resistant *Klebsiella* isolates are on the rise (Tzouvelekis *et al*, 2012; Olaitan *et al*, 2014b; Nation *et al*, 2015).

Several recent studies highlight the emergence of colistin resistance in MDR *K. pneumoniae* arising from loss-of-function mutations of the *mgrB* gene, a negative regulator of the PhoPQ signalling system (Lippa & Goulian, 2009; Cannatelli *et al*, 2013; Olaitan *et al*, 2014a; Poirel *et al*, 2015; Wright *et al*, 2015; Zowawi *et al*, 2015). The PhoPQ two-component system is a well-known regulator of envelope remodelling, chiefly the lipopolysaccharide (LPS) lipid A section, and contributes to bacterial resistance to innate immune killing (Groisman, 2001; Llobet *et al*, 2011). *K. pneumoniae* PhoPQ also governs lipid A plasticity *in vivo* and *in vitro* (Llobet *et al*, 2015) and plays a role in virulence as assayed using the wax moth *Galleria mellonella* infection model (Insua *et al*, 2013).

Alarmingly, the evidence indicates that *mgrB*-dependent colistin resistance is not associated with a significant fitness cost *in vitro* and is stably maintained in the absence of selective pressure

1 Centre for Experimental Medicine, Queen's University Belfast, Belfast, UK
2 School of Chemistry and Molecular Biosciences, The University of Queensland, Brisbane, Qld, Australia
3 Child Health Research Centre, The University of Queensland, Brisbane, Qld, Australia
*Corresponding author. Tel: +44 28 9097 6020; E-mail: j.bengoechea@qub.ac.uk; Twitter: @josebengoechea

 

(Cannatelli *et al*, 2015), which may explain the rapid dissemination of strains carrying this resistance mechanism in the clinical setting (Cannatelli *et al*, 2013, 2014; Olaitan *et al*, 2014a; Cheng *et al*, 2015; Poirel *et al*, 2015; Wright *et al*, 2015; Zowawi *et al*, 2015). However, the precise molecular resistance mechanisms of *mgrB*-associated colistin-resistant *K. pneumoniae* remain unclear. Moreover, it is currently unknown whether *mgrB* mutation confers any loss of virulence. This is particularly critical given the increasing number of *K. pneumoniae* infections caused by virulent clones and the ease with which *mgrB* mutations arise in the hospital setting.

To answer these clinically relevant questions, we constructed a *mgrB* mutant in a *K. pneumoniae* strain representative of the virulent *Klebsiella* clonal groups, therefore encoding the genetic determinants associated with severe human infections (Lery *et al*, 2014; Holt *et al*, 2015). The evidence presented in this work demonstrates that inactivation of *mgrB* not only results in colistin resistance, but also enhances *K. pneumoniae* virulence by decreasing the susceptibility to a plethora of antimicrobial peptides and attenuating the activation of host defence responses.

## Results

### Deletion of *mgrB* confers increased colistin resistance and multiple lipid A modifications in a PhoPQ-dependent manner

To solidify earlier reports indicating an association between *mgrB* gene mutation and colistin resistance, we constructed an *mgrB* mutant (referred to as 52145-Δ*mgrB* herein) in the wild-type virulent *K. pneumoniae* strain 52145 (Kp52145 herein). Control experiments showed that the growth kinetics in rich and minimal media were similar between the wild-type and 52145-Δ*mgrB* strains (Fig EV1A). The 52145-Δ*mgrB* mutant showed a slightly increased capacity to form a short-term biofilm compared to the wild type (Fig EV1B).

Upon Etest® minimal inhibitory concentration (MIC) testing, we demonstrated a 128-fold increase in colistin resistance in 52145-Δ*mgrB* (16.0 μg/ml) compared to the wild type (0.125 μg/ml; Fig 1A). Similar results were obtained when testing the susceptibility to polymyxin B (Appendix Fig S1), another cyclic polypeptide also used in clinical practice to treat MDR Gram-negative bacterial infections (Nation *et al*, 2015). When the *mgrB* mutant was complemented (strain 52145-Δ*mgrB*Com), MICs to colistin and polymyxin B were restored to wild-type levels (0.125 μg/ml; Fig 1A and Appendix Fig S1).

Resistance to colistin and polymyxin B is associated with remodelling in the lipid A (Olaitan *et al*, 2014b). To determine whether *mgrB* mutation results in lipid A changes, lipid A was extracted from the wild-type strain and the *mgrB* mutant using an ammonium hydroxide/isobutyric acid method and subjected to negative ion matrix-assisted laser desorption–ionisation time-of-flight (MALDI-TOF) mass spectrometry. Consistent with our earlier work, lipid A from the wild-type strain (Figs 1B and EV2, Appendix Table S1) showed hexa-acylated species (mass-to-charge ratio, [*m/z*] 1,824) corresponding to two glucosamines, two phosphates, four 3-OH-C$_{14}$ and two myristate (C$_{14}$), as well as two other peaks including *m/z* 1,840, corresponding to two glucosamines, two phosphates, four 3-OH-C$_{14}$, one C$_{14}$ and one -hydroxymyristate (C$_{14:OH}$), and *m/z*

2,063 consistent with the addition of palmitate (*m/z* 239) to the hexa-acylated (*m/z* 1,824) species to produce a hepta-acylated lipid A (Llobet *et al*, 2011, 2015). Lipid A isolated from 52145-Δ*mgrB* (Fig 1C) contained species *m/z* 1,824, *m/z* 1,840 and *m/z* 2,063 which were found also in the wild-type lipid A. In addition, we observed other lipid A species consistent with the addition of phosphoethanolamine (PEtN; *m/z* 124) and 4-amino-4-deoxy-L-arabinose (Ara4N; *m/z* 131) to the hexa-acylated *m/z* 1,824 to obtain species *m/z* 1,948 and *m/z* 1,955, respectively, and species *m/z* 2,079 consistent with the addition of palmitate to ion *m/z* 1,840 (Fig EV2). Other ions detected in the 52145-Δ*mgrB* lipid A comprised *m/z* 1,850 and *m/z* 1,866 consistent with four R-3-hydroxymyristoyl primary acyl chains, either one C$_{14}$ or one C$_{14:OH}$, respectively, and one palmitate (Llobet *et al*, 2015). Complementation of the *mgrB* mutant restored production of wild-type lipid A (Fig 1D).

We next investigated the lipid A of several previously published *K. pneumoniae* clinical strains linked to *mgrB* inactivation and colistin resistance (Poirel *et al*, 2015). We selected two isogenic ST-258 strains isolated from the same patient prior to (strain T1a) and after (strain T1b) colistin therapy and the development of *mgrB*-associated colistin resistance, along with four other colistin-resistant *mgrB* mutants (C21, C22, C2 and 15I5) from individual patients comprising various geographic sources, genetic backgrounds, multidrug resistance mechanisms and *mgrB* sequence mutations (Appendix Table S2). Appendix Fig S2 shows that the lipid A species produced by the T1a strain were similar to those observed in the wild-type Kp52145 strain, whereas T1b lipid A showed the same modifications as that of the 52145-Δ*mgrB* mutant. This latter finding was also consistent for each of the other clinical colistin-resistant *mgrB* mutant strains analysed.

Earlier work indicates that in *K. pneumoniae,* the dioxygenase LpxO is responsible for the generation of 2-hydroxymyristate, PagP is the acyltransferase required for the addition of palmitate to the lipid A, PmrC mediates the incorporation of PEtN (Llobet *et al*, 2011, 2015; Wright *et al*, 2015), whereas synthesis and addition of Ara4N is mediated by the *pmrHFIJKLM* operon (Llobet *et al*, 2011). To verify that these loci were indeed responsible for *mgrB*-dependent lipid A modifications, we constructed a range of double, triple, quadruple and quintuple *lpxO, pagP, pmrC* and *pmrF* mutants in the 52145-Δ*mgrB* background (Appendix Table S3) and analysed their lipid A structure. As anticipated, *lpxO* mutants lacked lipid A species containing 2-hydroxymyristate, *pagP* mutants did not contain species with palmitate, *pmrF* mutants lacked Ara4N-modified lipid A species, and *pmrC* mutants did not produce lipid A species modified with PEtN (Appendix Fig S3). It should be noted that the absence of a lipid A modification did not have an impact on the others.

The increased production of modified lipid A species in the *mgrB* mutant led us to investigate whether the expression of the loci responsible for these modifications was upregulated in the *mgrB* mutant background. To quantitatively assess the transcription of these loci, we used four transcriptional fusions containing a promoterless luciferase firefly gene (*lucFF*) under the control of the relevant locus promoter region. Each fusion (i.e. *lpxO::lucFF*, *pagP:: lucFF*, *pmrC::lucFF* and *pmrH::lucFF*) was introduced into Kp52145, 52145-Δ*mgrB* and 52145-Δ*mgrB*Com, and then, luciferase activity was measured. Compared to the wild-type strain, we observed significantly upregulated expression of all the transcriptional fusions

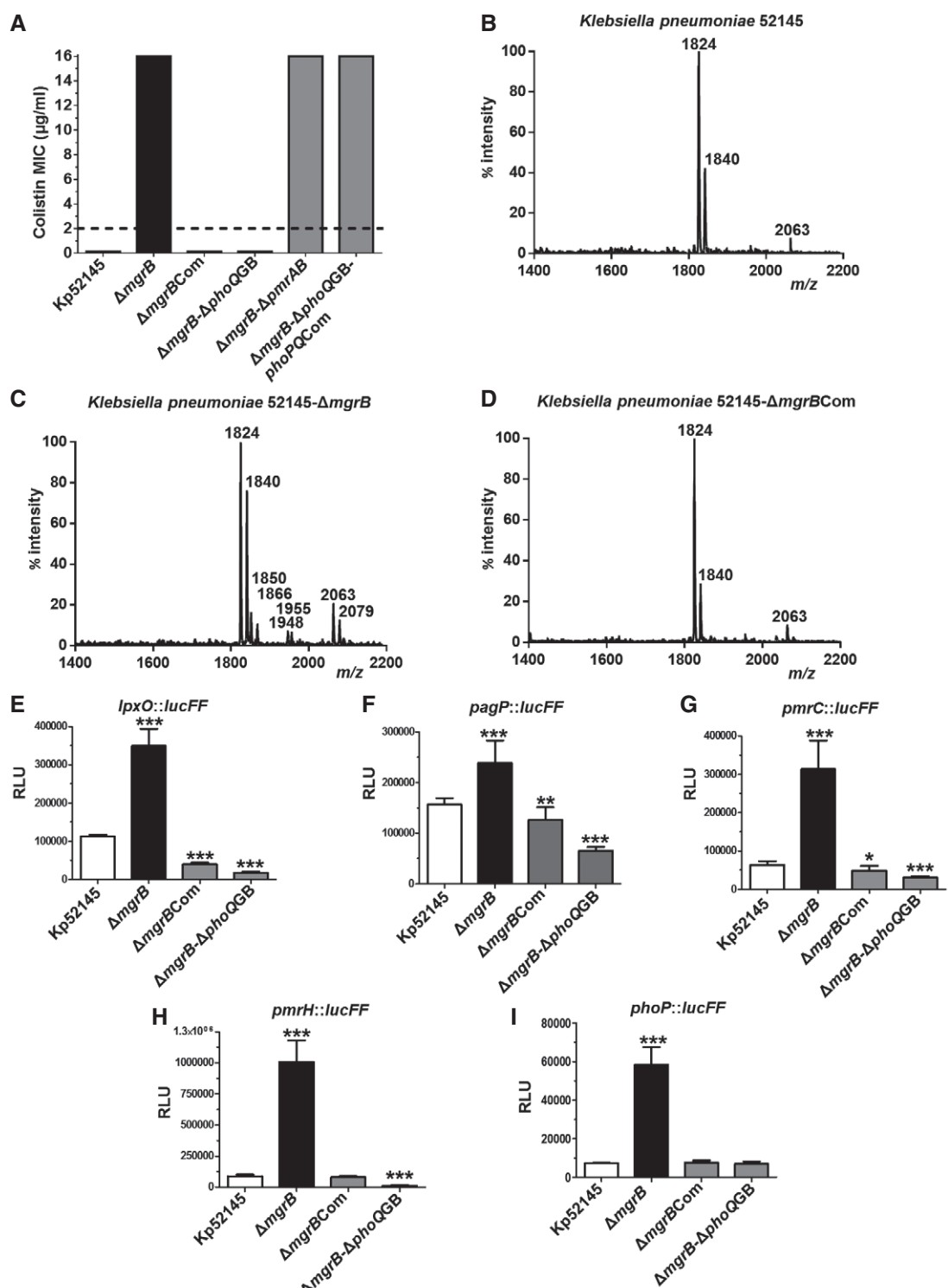

**Figure 1.  Deletion of *mgrB* in *K. pneumoniae* invokes polymyxin resistance and multiple lipid A modifications in a PhoPQ-dependent manner.**

A       Minimal inhibitory concentrations to colistin of the *K. pneumoniae* 52145, 52145-Δ*mgrB*, 52145-Δ*mgrB*Com, 52145-Δ*mgrB*-Δ*phoQ*GB, 52145-Δ*mgrB*-Δ*pmrAB* and
52145-Δ*mgrB*-Δ*phoQ*GB-*phoPQ*Com strains. The broken line represents the European Committee on Antimicrobial Susceptibility Testing MIC breakpoint.

B–D     Negative ion MALDI-TOF mass spectrometry spectra of lipid A purified from (B) *K. pneumoniae* 52145, (C) 52145-Δ*mgrB* and (D) 52145-Δ*mgrB*Com strains. Data
represent the mass-to-charge (*m/z*) ratios of each lipid A species detected and are representative of three extractions.

E–I     Activity of the *lpxO* (E), *pagP* (F), *pmrC* (G) and *pmrH* (H) and *phoP* (I) promoters in *K. pneumoniae* 52145, 52145-Δ*mgrB*, 52145-Δ*mgrB*Com, 52145-Δ*mgrB*-Δ*phoQ*GB
carrying *lucFF* transcriptional fusions. Values (expressed in relative luminescence units) are presented as the mean ± SD of three independent experiments
measured in triplicate. ***$P < 0.0005$; **$P = 0.0071$; *$P = 0.019$; versus 52145 determined using two-way unpaired *t*-test.

in 52145-Δ*mgrB* (Fig 1E–H). In contrast, the luciferase activities of all transcriptional fusions in the *mgrB*-complemented strain were not significantly different than those observed in the wild-type background.

Based on the fact that the two-component PhoPQ and PmrAB systems govern the remodelling of *Klebsiella* lipid A (Llobet *et al*, 2011), we sought to determine the contribution of these two-component systems to *mgrB*-mediated lipid A changes. Lipid A produced by the *mgrB-phoQ* and *mgrB-phoQ-pmrAB* mutants resembled that of the wild type, whereas lipid A synthesised by the *mgrB-pmrAB* mutant and Δ*mgrB*-Δ*phoQ* complemented with *phoPQ* (52145-Δ*mgrB*-Δ*phoQ*GB-*phoPQ*Com) were similar to the lipid A produced by the *mgrB* mutant (Fig EV3 and Appendix Table S1). Taken together, these data suggest that *mgrB*-mediated lipid A modifications are PhoPQ dependent. Further stressing the connection between MgrB and PhoPQ, *phoP::lucFF* activity was higher in the *mgrB* mutant than in the wild-type strain (Fig 1I), and the luciferase activities of *lpxO*, *pagP*, *pmrF* and *pmrC* transcriptional fusions were not significantly different between the *mgrB-phoQ* mutant and the wild type (Fig 1E–I). Importantly, we also confirmed that the colistin and polymyxin B resistance phenotype observed in the *mgrB* mutant was PhoPQ but not PmrAB dependent (Fig 1A and Appendix Fig S1).

Collectively, these data demonstrate that deletion of the *mgrB* gene gives rise to colistin and polymyxin B resistance in a PhoPQ-dependent manner. Upregulation of the PhoPQ system results in increased expression of *lpxO*, *pagP*, *pmrC* and *pmrF*, which in turn facilitates lipid A modifications with 2-hydroxymyristate, Ara4N, PEtN and palmitate. Significantly, these lipid A modifications were also observed in a broad selection of other clinically derived *mgrB* mutants showing colistin-resistant phenotypes.

### Inactivation of *mgrB* mediates increased resistance to antimicrobial peptides

It is widely recognised that polymyxin antibiotics share a similar mode of action to cationic antimicrobial peptides (AMPs) with various studies demonstrating the relationship between polymyxin and AMP resistance (Groisman, 2001; McPhee *et al*, 2003; Campos *et al*, 2004). Therefore, we speculated that 52145-Δ*mgrB* may also show decreased susceptibility to AMPs. To test this hypothesis, we exposed Kp52145, 52145-Δ*mgrB* and 52145-Δ*mgrB*Com to four different human AMPs over 1 h and determined the proportion of surviving organisms. For human neutrophil peptide-1 (1.2 μM), we showed significantly ($P = 0.0006$) increased resistance in 52145-Δ*mgrB* when compared to the wild type (Fig 2A). Likewise, there was significantly ($P \leq 0.013$) increased survival of 52145-Δ*mgrB* following exposure to each of three β-defensins (Fig 2B–D). The complemented strain showed similar AMP susceptibility to that of the wild type, indicating that *mgrB* mutation confers protection against human AMPs.

*In vivo* it is appreciated that several AMPs act in a synergistic manner in the infected tissue to combat invading pathogens (Afacan *et al*, 2013). Therefore, exposure to one AMP does not recapitulate the AMP challenge faced by a pathogen *in vivo*. To mimic this *in vivo* scenario, we thus turned to the *Galleria mellonella* infection model where only hours after infection, multiple AMPs are synthesised and released into the hemolymph to neutralise bacterial

infection (Kavanagh & Reeves, 2004; Insua *et al*, 2013). Haemolymph was collected from *G. mellonella* challenged with heat-killed (HK) *Escherichia coli,* and the susceptibility of Kp52145, 52145-Δ*mgrB* and 52145-Δ*mgrB*Com to the AMPs present in the haemolymph was determined using a radial diffusion bioassay. The *mgrB* mutant was significantly more resistant than the wild type to AMPs present in *G. mellonella* haemolymph (Fig 2E). Complementation restored the *mgrB* mutant susceptibility to wild-type levels, demonstrating that *mgrB* mutation confers resistance to a repertoire of AMPs produced in response to bacterial infections.

### Multiple lipid A modifications contribute to *mgrB*-mediated colistin resistance

To delineate the relative contributions of the lipid A modifications to the polymyxin resistance phenotype of the *mgrB* mutant, we undertook susceptibility testing using the Etest® on each of the double-, triple- and quadruple-mutant Kp52145 strains. We observed a substantial reduction in MIC for both colistin and polymyxin B only in *lpxO* and/or *pmrF* mutant strains (Fig 3A and B).

To provide additional evidence for the involvement of these lipid A modifications in the *mgrB*-mediated resistance phenotype, we then calculated the per cent survival of the wild-type and mutant strains after a 1-h colistin (20 μg/ml) challenge. As anticipated, we observed a marked difference ($P < 0.0001$) in mean per cent survival between the wild-type ($3.8 \pm 3.8\%$) and 52145-Δ*mgrB* ($83.6 \pm 15.34\%$) strains (Fig 3C). Complementation fully restored the colistin resistance of the *mgrB* mutant to wild-type levels (Fig 3C), with similar results observed for polymyxin B (Fig 3D). Extended analyses of the double, triple and quadruple mutants when exposed to colistin provided confirmation of the results obtained using the Etest®. Indeed, compared to 52145-Δ*mgrB,* the 52145-Δ*mgrB*-Δ*lpxO* and 52145-Δ*mgrB*-Δ*pmrF* strains showed a significant ($P \leq 0.02$) reduction in mean per cent survival (Fig 3C). The contribution of lipid A modifications with Ara4N and 2-hydroxymyristate to survival was also confirmed in the 52145-Δ*pmrC*-Δ*lpxO*-Δ*mgrB*, 52145-Δ*mgrB*-Δ*lpxO*-Δ*pmrF* and 52145-Δ*pmrC*-Δ*lpxO*-Δ*mgrB*-Δ*pmrF* strains. In accordance with the colistin Etest®, we observed no difference in per cent survival between 52145-Δ*pagP*-Δ*mgrB* and 52145-Δ*mgrB* ($P = 0.99$), which corroborated earlier work showing that lipid A modification with palmitate is not involved in *K. pneumoniae* polymyxin resistance (Llobet *et al*, 2011). In contrast to the Etest® MIC results, our data did reveal a significant difference in survival between 52145-Δ*pmrC*-Δ*mgrB* and 52145-Δ*mgrB* ($P < 0.001$); but with the 52145-Δ*pmrC*-Δ*lpxO*-Δ*mgrB* triple mutant, the anticipated amplified effect of removing both *pmrC* and *lpxO* was not evident.

In summary, we provide definitive evidence that the colistin and polymyxin B resistance phenotype of 52145-Δ*mgrB* occurs through at least two key modifications to the lipid A structure: the additions of Ara4N and 2-hydroxymyristate.

### Hypervirulence of *K. pneumoniae* 52145-Δ*mgrB* in the *G. mellonella* infection model

A strong correlation between the virulence of several MDR bacteria, including *K. pneumoniae*, in *G. mellonella* and mammalian virulence models has been established (Jander *et al*, 2000; Insua *et al*,

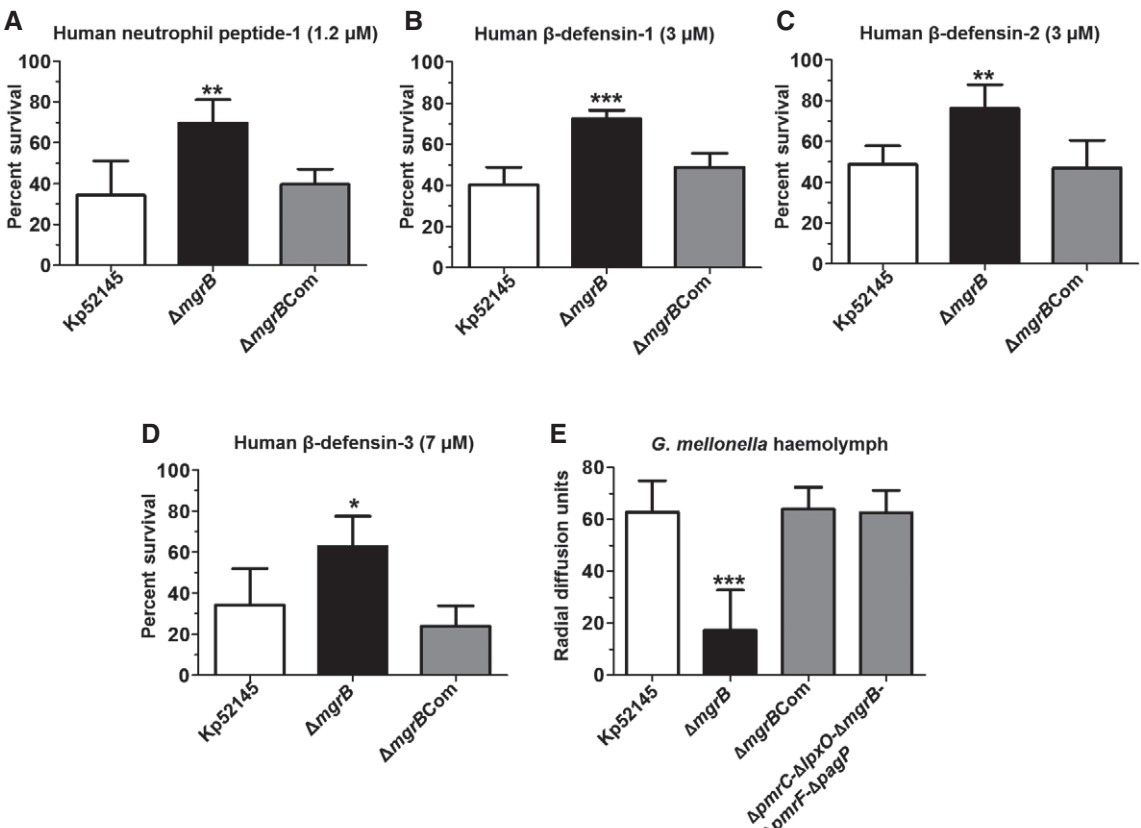

**Figure 2.** **Deletion of *mgrB* in *K. pneumoniae* increases resistance to antimicrobial peptides.**

A–D Per cent survival of *K. pneumoniae* 52145, 52145-Δ*mgrB* and 52145-Δ*mgrB*Com following 1-h exposure to the following: (A) human neutrophil peptide-1 (1.2 μM), (B) human β-defensin-1 (3 μM), (C) human β-defensin-2 (3 μM) and (D) human β-defensin-3 (7 μM). Values are presented as the mean ± SD of three independent experiments measured in duplicate. ***$P$ = 0.0006; **$P$ = 0.01–0.001; *$P$ = 0.013; versus Kp52145 determined using two-way unpaired *t*-test.

E Resistance of *K. pneumoniae* 52145, 52145-Δ*mgrB* and 52145-Δ*mgrB*Com to antimicrobial factors produced by *G. mellonella* after 24-h exposure to $10^6$ heat-killed *E. coli* cells. The experiments were undertaken using a radial diffusion bioassay with data expressed as radial diffusion units (10 units = 1 mm). Values are presented as the mean ± SD of three independent experiments measured in triplicate. ***$P$ < 0.0001; versus 52145 determined using two-way unpaired *t*-test.

2013; Jacobs *et al*, 2014). Importantly, we demonstrated that the *G. mellonella* model recapitulates key features of *K. pneumoniae* infection biology (Insua *et al*, 2013). To determine the pathogenic potential of the *mgrB* mutant in *G. mellonella*, we injected an equivalent dose of the wild-type, 52145-Δ*mgrB*, 52145-Δ*mgrB*-Δ*phoQ*GB and complemented strains into *G. mellonella* and monitored bacterial killing over time. No mortality was observed in the control phosphate-buffered saline (PBS)-injected *G. mellonella* larvae. After 72 h, 60–70% of the larvae challenged with Kp52145 and 52145-Δ*mgrB*-Δ*phoQ*GB survived. In stark contrast, only 18% of the larvae infected with 52145-Δ*mgrB* survived (Fig 4A). Complementation of the *mgrB* mutant restored the virulence to wild-type levels, indicating that *mgrB* mutation mediates the hypervirulence phenotype. This increase in virulence was also observed when the killing potential of a clinical colistin-resistant *mgrB* isolate was investigated. Figure 4B shows that infection with the colistin-resistant *mgrB* mutant (T1b) strain resulted in increased mortality as compared to the isogenic colistin-susceptible ancestral (T1a) strain.

It has been previously shown that boosting immunity in *G. mellonella* by pre-immunisation with HK-*E. coli* provides

protection against subsequent *K. pneumoniae* infection (Insua *et al*, 2013). Therefore, we sought to determine whether *mgrB* mutation also confers hypervirulence in *G. mellonella* with stimulated host immunity. To this end, we inoculated larvae with $10^6$ HK-*E. coli* and then after 24 h infected with $10^6$ CFUs of Kp52145 or 52145-Δ*mgrB*. Confirming earlier work, larvae infected with Kp52145 showed a significant survival improvement if pre-immunised (Fig 4D). However, there was no survival difference between the 52145-Δ*mgrB*-infected larvae receiving pre-inoculation with HK-*E. coli* and the PBS vehicle ($P$ = 0.76; Fig 4D), thus indicating that *mgrB* mutation increases virulence even in the pre-immunised host.

Earlier work from our group illustrated that *K. pneumoniae* capsule polysaccharide (CPS) is crucial for virulence in *G. mellonella* (Insua *et al*, 2013). To explore the contribution of CPS to the hypervirulence of the *mgrB* mutant, we compared the killing ability of the *cps* mutant (52145-Δ*manC*) and the double *mgrB-cps* mutant (52145-Δ*mgrB*-Δ*manC*). One hundred per cent of the larvae infected with $10^6$ CFUs of either 52145-Δ*manC* or 52145-Δ*mgrB*-Δ*manC* survived after 72 h (Fig EV4A). Control experiments

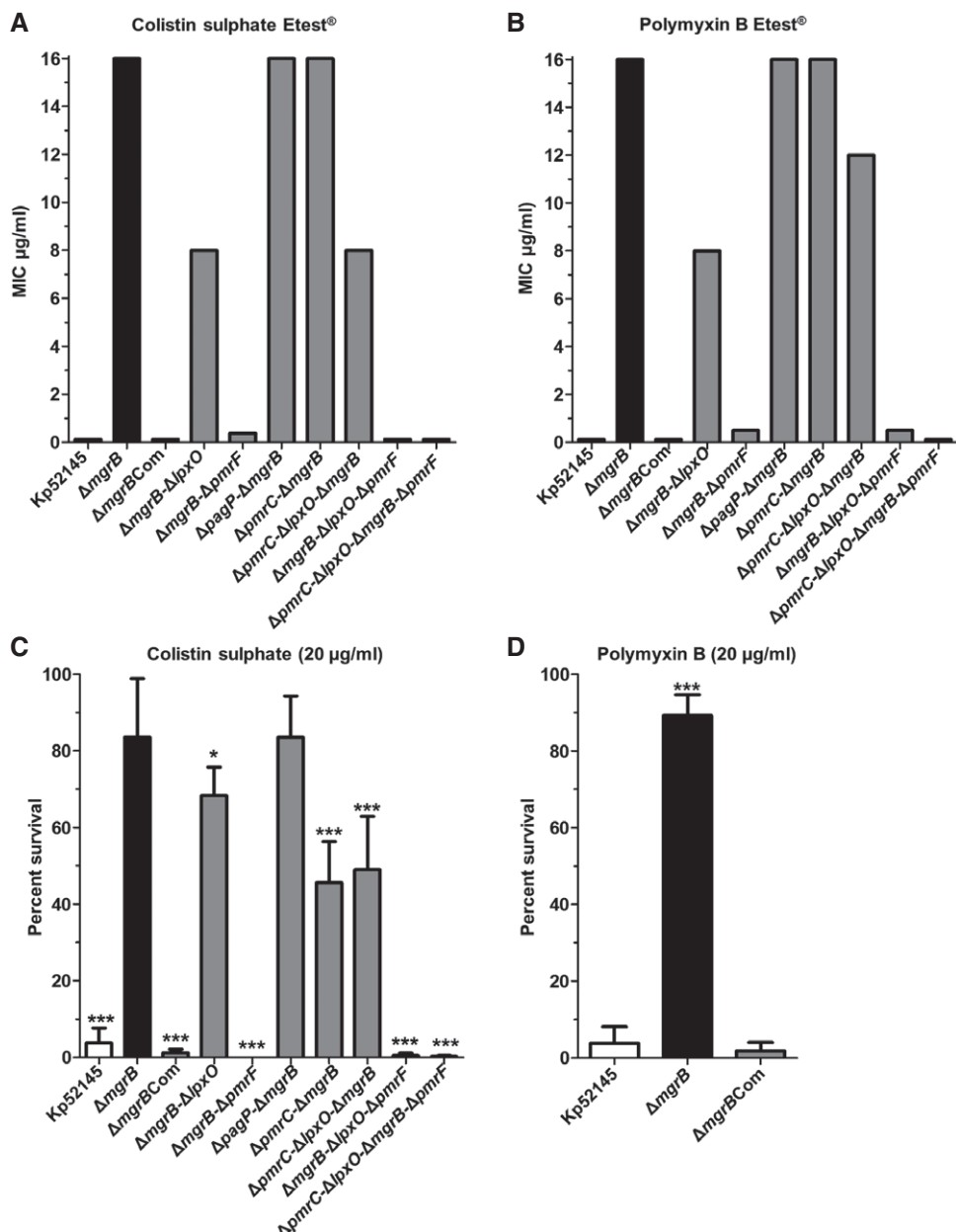

**Figure 3.  Effect of *lpxO*, *pmrF*, *pagP* and *pmrC* mutation on the polymyxin resistance of *K. pneumoniae mgrB* mutant.**

A, B    Etest® minimal inhibitory concentrations to colistin and polymyxin B of *K. pneumoniae* 52145, 52145-Δ*mgrB*, 52145-Δ*mgrB*Com compared to the double, triple and quadruple 52145-Δ*mgrB* mutant strains.

C, D    Per cent survival of the *K. pneumoniae* 52145 wild-type and mutant strains after exposure to 20 µg/ml colistin (C) and polymyxin B (D) over 1 h. Values are presented as the mean ± SD of three independent experiments measured in duplicate. ***$P < 0.0001$; *$P = 0.0195$; versus 52145-Δ*mgrB* determined using a two-way unpaired *t*-test.

showed that both the wild type and *mgrB* mutant expressed the same amount of cell-bound CPS (187.1 ± 8.8 µg/10$^9$ CFUs versus 177.7 ± 14.2 µg/10$^9$ CFUs, respectively; $P = 0.30$). Nor was there any difference in the *cps::lucFF* activity determined in Kp52145 and 52145-Δ*mgrB* backgrounds (Fig EV4B). Overall, these data demonstrate that both *mgrB* mutation and CPS are necessary to increase *K. pneumoniae* virulence in *G. mellonella*, although MgrB does not control CPS expression.

To provide mechanistic insights into the *mgrB* hypervirulence phenotype, we investigated whether any of the lipid modifications found in the 52145-Δ*mgrB* lipid A contributed to the heightened virulence. Upon infection with the double-, triple- and quadruple-mutant strains, there was no difference in mortality when compared to 52145-Δ*mgrB* (Appendix Fig S4). However, infection with the mutant lacking all four lipid A modifications, strain 52145-Δ*pmrC*-Δ*lpxO*-Δ*mgrB*-Δ*pmrF*-Δ*pagP*, revealed a significantly different per

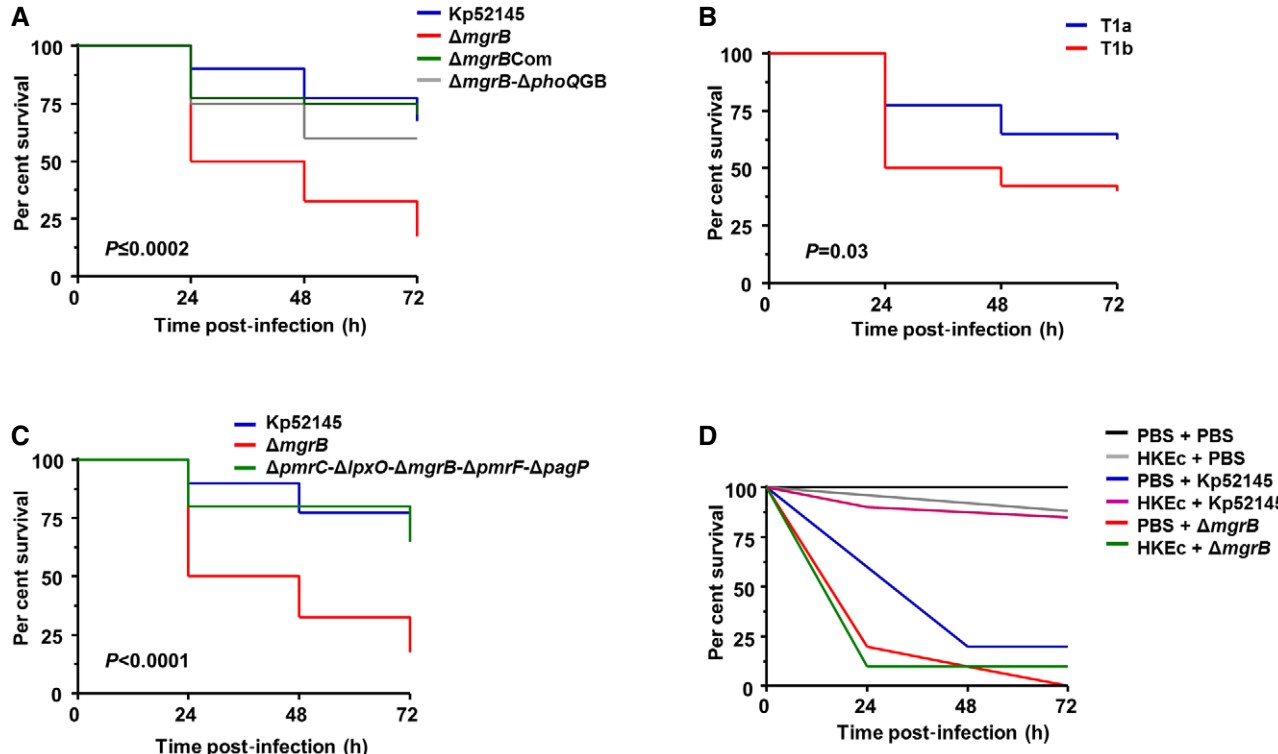

**Figure 4. *K. pneumoniae mgrB* mutant displays increased virulence in the *G. mellonella* waxworm infection model.**

A–C   Kaplan–Meier plots showing the per cent survival of *G. mellonella* over 72 h post-infection with $10^5$ organisms of the following: (A) *K. pneumoniae* 52145 (blue), 52145-Δ*mgrB* (red), 52145-Δ*mgrB*Com (green) and 52145-Δ*mgrB*-Δ*phoQ*GB (grey), (B) clinical *K. pneumoniae* strains T1a (blue) and T1b (red), and (C) *K. pneumoniae* 52145 (blue), 52145-Δ*mgrB* (red) and 52145-Δ*pmrC*-Δ*lpxO*-Δ*mgrB*-Δ*pmrF*-Δ*pagP* quintuple-mutant (green) strains. Forty larvae were infected in each group. Level of significance was determined using the log-rank (Mantel–Cox) test with Bonferroni correction for multiple comparisons where applicable [α = (A) 0.0008; (B) 0.05; (C) 0.017]. *P*-values presented correspond to the difference between (A) 52145-Δ*mgrB* and the other 52145 strains and (B) T1a versus T1b.

D     Survival curve of *G. mellonella* primed with $10^6$ heat-killed *E. coli* cells (HKEc) or phosphate-buffered saline (PBS) and then exposed to PBS (black, grey) or $10^6$ organisms of *K. pneumoniae* 52145 (blue, magenta) and 52145-Δ*mgrB* (red, green) over 72 h. Ten larvae were infected in each group. Level of significance was determined using the log-rank (Mantel–Cox) test with Bonferroni correction for multiple comparisons (α = 0.008).

cent survival compared to 52145-Δ*mgrB* (Fig 4C). Furthermore, there was no difference in mortality when compared to that triggered by the wild type. Collectively, these data provide evidence that 52145-Δ*mgrB* pathogenicity in *G. mellonella* is combinatorial with all four lipid A modifications required to result in the overall virulence phenotype.

### *K. pneumoniae* 52145-Δ*mgrB* infection is not attenuated in a murine infection model

The hypervirulence of 52145-Δ*mgrB* in *G. mellonella* prompted us to examine the ability of this mutant to cause infection in a mammalian infection model. C57BL/6 mice were intranasally inoculated with $3 \times 10^5$ organisms, and bacterial loads in lung, spleen and nasal-associated lymphoid tissue (NALT) homogenates were determined at 24 h post-infection. Bacterial loads of 52145-Δ*mgrB* in the NALT, lung and splenic samples were similar to those of the wild type ($P \geq 0.14$; Fig 5A–C). We also sought to compare the level of inflammatory cytokine and antimicrobial peptide expression in the lungs of mice infected with Kp52145 and 52145-Δ*mgrB*. Expression of *Tnfa*, *Ifnb*, *Il1b*, *Il6*, *Il10* or *Il12* was measured by real-time quantitative PCR (RT-qPCR) with

levels of all cytokines, excluding *Ifnb*, higher in lungs of infected mice than in lungs of non-infected animals (Appendix Fig S5; $P < 0.05$ for all comparisons versus non-infected mice). Interestingly, the *mgrB* mutant induced the same levels of all cytokines as the wild-type strain, and this was also true when the expression of murine defensins was analysed (Appendix Fig S6). Overall, these findings suggest that *mgrB* mutation does not compromise *K. pneumoniae* immune-evasion strategies in the mouse model.

### *K. pneumoniae* 52145-Δ*mgrB* attenuates the expression of antimicrobial peptides in the *G. mellonella* infection model

We have previously demonstrated that there is a correlation between *K. pneumoniae* virulence and the expression of *G. mellonella* AMPs. The levels of AMPs are higher in larvae infected with an avirulent *K. pneumoniae cps* mutant than in larvae infected with the wild-type strain (Insua *et al*, 2013). The increased virulence of the *mgrB* mutant led us to investigate whether this mutant affects the expression of AMPs in *G. mellonella*. After 8 h of infection, the expression of lysozyme, gallerimycin and galiomycin was significantly ($P < 0.0001$) lower in 52145-Δ*mgrB*-infected larvae

than in the wild-type-infected larvae (Fig 5D and Appendix Fig S7). Infection with the 52145-Δ*pmrC*-Δ*lpxO*-Δ*mgrB*, 52145-Δ*mgrB*-Δ*lpxO*-Δ*pmrF* and 52145-Δ*pmrC*-Δ*lpxO*-Δ*mgrB*-Δ*pmrF* triple and quadruple mutants restored lysozyme expression to wild-type levels (Fig 3D). Based on these data, we surmised that the *lpxO* mutation present in each of these three strains played a primary role in this phenotype. To explore this hypothesis, lysozyme expression was assessed during infections with an *lpxO* mutant (52145-Δ*lpxO*), the 52145-Δ*mgrB*-Δ*lpxO* double mutant and 52145-Δ*mgrB*-Δ*lpxO* complemented with *lpxO* (52145-Δ*mgrB*-Δ*lpxO*-*lpxO*Com). Interestingly, expression of this AMP among the single and double *Klebsiella* mutant-infected larvae was similar to the wild type ($P \geq 0.38$); however, the 52145-Δ*mgrB*-Δ*lpxO*-*lpxO*Com strain induced an identical phenotype to the 52145-Δ*mgrB* mutant ($P < 0.0001$; Fig 5D). Likewise, expression of gallerimycin and galiomycin was reduced among larvae infected with 52145-Δ*mgrB*-Δ*lpxO*-*lpxO*Com (Fig EV5). These findings demonstrated that *mgrB* mutation is associated with an attenuated expression of *G. mellonella* AMPs upon infection. Furthermore, this phenotype is dependent on LpxO-controlled lipid A modification.

### Early inflammatory responses in macrophages are subdued upon infection with 52145-Δ*mgrB*

The recognition of the lipid A pattern by the TLR4/MD-2 complex activates NF-κB and MAP kinase (MAPK)-regulated defence responses necessary to clear infections. The distinct lipid A produced by the *mgrB* mutant and the *mgrB*-dependent attenuation of *G. mellonella* defence responses led us to investigate the signalling pathways and inflammatory responses triggered by the *mgrB* mutant in macrophages. Immortalised bone marrow-derived macrophages (iBMDMs) were infected with the wild type, *mgrB* mutant and the *mgrB* mutant-complemented strain with the activation of NF-κB and MAPKs assessed by immunoblotting. In the canonical NF-κB activation pathway, nuclear translocation of NF-κB is preceded by phosphorylation and subsequent degradation of IκBα. All strains triggered the degradation of IκBα (Fig 6A). Whereas the wild type and *mgrB*-complemented strain induced phosphorylation of the p38, JNK and ERK MAPKs, the *mgrB* mutant triggered reduced phosphorylation of the three MAPKs (Fig 6B). Similar results were obtained when MH-S macrophages were infected (Appendix Fig S8A), thus indicating that the failure to activate MAPKs by the *mgrB* mutant is not cell type dependent. Control experiments showed that *K. pneumoniae*-triggered TNF-α is dependent on MAPKs JNK and ERK but not on p38 because the levels of this cytokine were significantly lower only in the supernatants of infected macrophages treated with the ERK and JNK inhibitors (U0126 and SP600125, respectively) than in infected cells treated with vehicle solution or the p38 inhibitor (Appendix Fig S8B). As anticipated, based on the reduced activation of JNK and ERK by the *mgrB* mutant, levels of TNF-α in the supernatants of macrophages infected with the *mgrB* mutant were significantly lower than those found in the supernatants of cells infected with either the wild type or the *mgrB*-complemented strain. Mechanistically, the reduced activation of MAPKs JNK and ERK by the *mgrB* mutant was dependent on the four *mgrB*-controlled lipid A modifications because only the quintuple mutant triggered a consistently similar pattern of MAPK phosphorylation as the wild-type strain (Fig 6C). In good agreement

with these results, the quintuple mutant induced levels of TNF-α similar to the wild-type strain (Fig 6D). Taken together, our data demonstrate that inactivation of *mgrB* in *K. pneumoniae* results in reduced activation of inflammatory signalling pathways in macrophages.

## Discussion

The increasing isolation of MDR Gram-negative pathogens resistant to polymyxins, which are considered a last treatment option for these infections, is a health challenge worldwide. It is therefore important to define the molecular mechanisms responsible for polymyxin resistance and also to address whether the virulence of the pathogen is affected. The latter aspect is often neglected in those studies investigating MDR pathogens, and it is crucially important in the clinical setting where infections occur in immunocompromised patients.

Several studies have reported the emergence of colistin resistance in MDR *K. pneumoniae* arising from loss-of-function mutations of the *mgrB* gene (Cannatelli *et al*, 2013, 2014; Lopez-Camacho *et al*, 2014; Olaitan *et al*, 2014a; Cheng *et al*, 2015; Giani *et al*, 2015; Poirel *et al*, 2015; Wright *et al*, 2015; Zowawi *et al*, 2015). In this work, by combining biochemistry and genetics, we demonstrate that *mgrB* mutation in *K. pneumoniae* is associated with PhoPQ-governed lipid A remodelling which confers resistance to polymyxins and mammalian antimicrobial peptides. Mechanistically, our findings reveal that the lipid A modifications with Ara4N and 2-hydroxymyristate mediate the resistance to polymyxins. Our results revealed that *mgrB* mutation increases *K. pneumoniae* virulence in the *G. mellonella* infection model, whereas it does not compromise *K. pneumoniae* survival in the mouse pneumonia model, hence further reinforcing the notion that the development of antibiotic resistance is not inexorably linked to decreased virulence and fitness costs.

Earlier work showed that MgrB is a small (47 amino acids), membrane-bound peptide which acts as negative feedback regulator of the PhoPQ two-component regulatory system (Lippa & Goulian, 2009). The fact that PhoPQ controls the expression of loci required for remodelling of the lipid A led to the assumption that *mgrB* mutation should be associated with changes in the lipid A structure, chiefly the addition of Ara4N to the lipid A. This notion was based on indirect experimental evidence showing that the transcription of the *pmrF* operon, responsible for the addition of Ara4N to the lipid A, is upregulated in an *mgrB* mutant background (Cannatelli *et al*, 2013; Wright *et al*, 2015). However, none of these assumptions have been formally proven. In this work, we demonstrate extensive remodelling of the lipid A produced by *K. pneumoniae mgrB* mutants, which is modified with Ara4N, 2-hydroxymyristate, palmitate and PEtN. Mechanistically, we have demonstrated that the *pmrF* operon, *lpxO*, *pagP* and *pmrC* are the loci responsible for these lipid A modifications in the *mgrB* mutant. Furthermore, we provide conclusive evidence showing the crucial role of PhoPQ governing all of these lipid A modifications. PhoPQ is responsible for the upregulated expression of the *pmrF* operon, *lpxO*, *pagP* and *pmrC* observed in the *mgrB* mutant. However, our analysis ruled out the contribution of the PmrAB two-component system despite its previously described

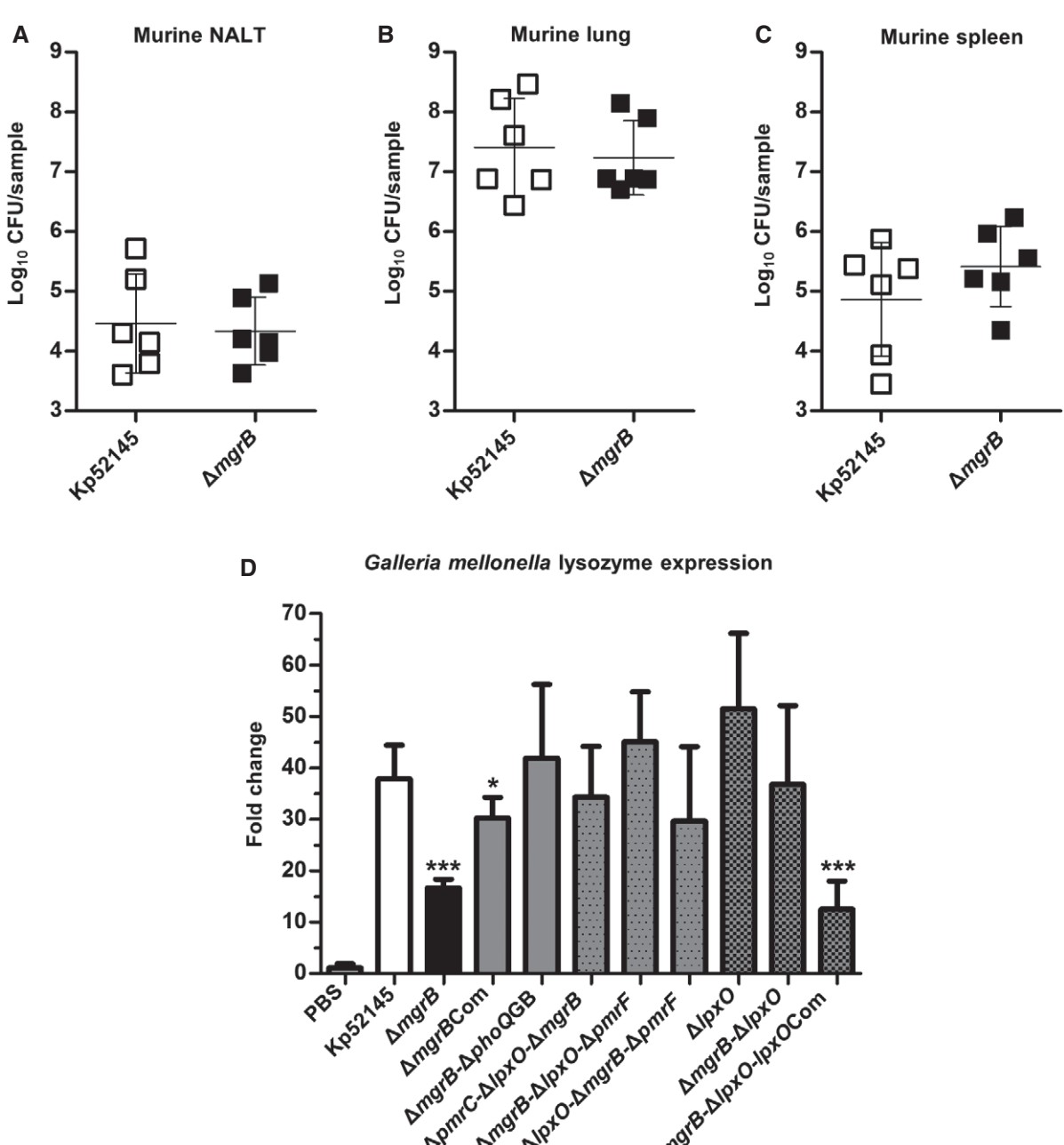

**Figure 5. Virulence of *K. pneumoniae mgrB* mutant in a murine intranasal infection model and expression of lysozyme in *G. mellonella* upon infection.**

A–C Bacterial loads of *K. pneumoniae* 52145 and 52145-Δ*mgrB* in nasal-associated lymphoid tissue (NALT), lung and spleen homogenates of infected mice after 24 h. Six mice per group were infected with $\log_{10}$ colony-forming unit values presented as the mean ± SD. Data were analysed using two-way unpaired *t*-test.

D Expression of lysozyme produced by *G. mellonella* after 8 h of infection with *K. pneumoniae* 52145, 52145-Δ*mgrB*, 52145-Δ*mgrB*Com, 52145-Δ*mgrB*-Δ*phoQ*GB, 52145-Δ*pmrC*-Δ*lpxO*-Δ*mgrB*, 52145-Δ*mgrB*-Δ*lpxO*-Δ*pmrF*, 52145-Δ*pmrC*-Δ*lpxO*-Δ*mgrB*-Δ*pmrF*, 52145-Δ*lpxO*, 52145-Δ*mgrB*-Δ*lpxO* and 52145-Δ*mgrB*-Δ*lpxO*-*lpxO*Com as determined by reverse transcriptase quantitative real-time PCR. Three larvae per group were infected, and values are presented as the mean ± SD of two independent cDNA preparations measured in duplicate. ***$P < 0.0001$; *$P = 0.035$; versus 52145 determined using two-way unpaired *t*-test.

contribution controlling some of these lipid A modifications in *K. pneumoniae* and other bacteria (Gunn *et al*, 1998; Groisman, 2001; Mitrophanov *et al*, 2008; Arroyo *et al*, 2011).

By testing a panel of mutants constructed in the *mgrB* mutant, we uncovered that polymyxin resistance is mostly dependent on the lipid A modification with Ara4N and 2-hydroxymyristate, although

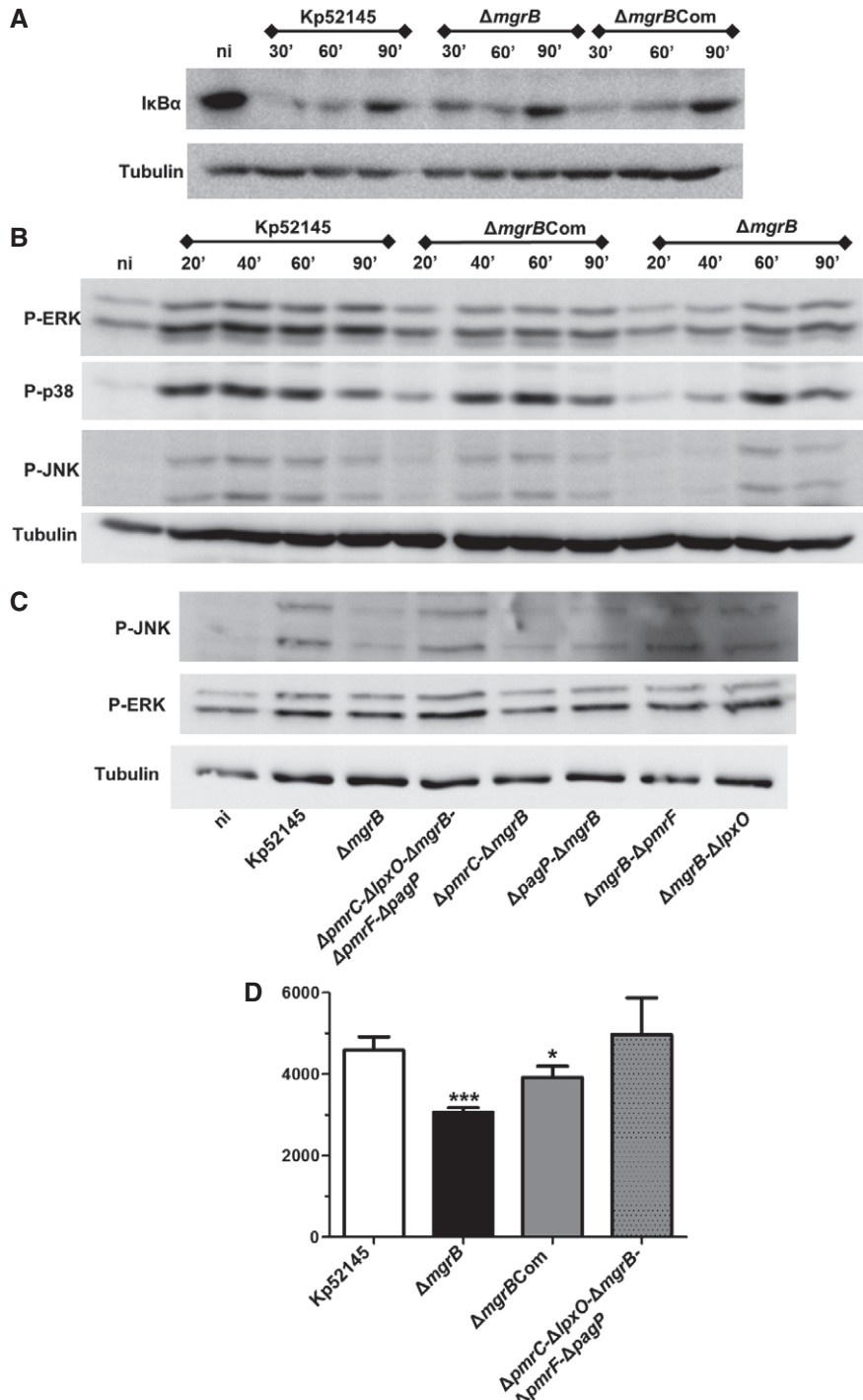

**Figure 6.    *mgrB* inactivation results in downregulation of early inflammatory responses in macrophages upon infection.**

A    Immunoblot analysis of IκBα and tubulin levels in lysates of iBMDM cells infected with *K. pneumoniae* 52145, 52145-Δ*mgrB* and 52145-Δ*mgrB*Com for the indicated times.

B    Immunoblot analysis of phospho-ERK (P-ERK), phospho-p38 (P-p38), phospho-JNK (P-JNK) and tubulin levels in lysates of iBMDMs cells infected with *K. pneumoniae* 52145, 52145-Δ*mgrB*, 52145-Δ*mgrB*Com and for the indicated times.

C    Immunoblot analysis of phospho-ERK (P-ERK), phospho-JNK (P-JNK) and tubulin levels in lysates of iBMDMs cells infected with *K. pneumoniae* 52145, 52145-Δ*mgrB*, 52145-Δ*pmrC*-Δ*lpxO*-Δ*mgrB*-Δ*pmrF*-Δ*pagP*, 52145-Δ*mgrB*-Δ*pmrC*, 52145-Δ*mgrB*-Δ*pagP*, 52145-Δ*mgrB*-Δ*pmrF* and 52145-Δ*mgrB*-Δ*lpxO* for 40 min.

D    TNF-α secretion by iBMDM macrophages stimulated for 6 h with $1 \times 10^5$ UV-killed *K. pneumoniae* 52145, 52145-Δ*mgrB*, 52145-Δ*mgrB*Com and 52145-Δ*pmrC*-Δ*lpxO*-Δ*mgrB*-Δ*pmrF*-Δ*pagP*. \*\*\**P* < 0.0001; \**P* = 0.02; versus 52145 determined using two-way unpaired *t*-test (mean ± SD).

Data information: Data are representative of at least three independent experiments.
Source data are available online for this figure.

a contribution of the PmrC-dependent PEtN modification was observed when the polymyxin susceptibility was assessed using a standard assay in the antimicrobial peptide field. Whereas it is well established that the substitution of the lipid A phosphates with Ara4N or PEtN limits the interaction of polymyxins with the bacterial surface, thereby mediating resistance (Olaitan *et al*, 2014b), the role of 2-hydroxymyristate remains unknown. Nonetheless, it is becoming evident that the presence of hydroxyl groups in the lipid A provides a competitive advantage in the presence of antimicrobial peptides (Hankins *et al*, 2011; Llobet *et al*, 2015). The fact that other Gram-negative pathogens (*Salmonella, Legionella, Acinetobacter, Pseudomonas)* also synthesise lipid A species possessing a hydroxyl group on a secondary acyl chain (Kulshin *et al*, 1991; Zahringer *et al*, 1995; Gibbons *et al*, 2000; Beceiro *et al*, 2011) could indicate that this lipid A modification is a conserved mechanism to counteract antimicrobial peptides. It is beyond the scope of this work to precisely define the molecular mechanism underlying 2-hydroxymyristate-dependent colistin resistance. However, it has been argued by Nikaido and co-workers that LpxO-dependent lipid A modification may increase the H bonding between neighbouring LPS molecules, thereby limiting polymyxins-triggered self-promoted pathway (Hancock *et al*, 1981; Nikaido, 2003).

Our work also highlights a connection between virulence and antimicrobial resistance. *mgrB* mutants were hypervirulent in the *G. mellonella* infection model even if the immunity of larvae was boosted prior to infection. Recently, we reported that following innate immune induction, the haemolymph of *G. mellonella* contains antimicrobials which inhibited *K. pneumoniae* growth (Insua *et al*, 2013). It was then plausible to speculate that the heightened virulence of the *mgrB* mutant was due to its increased resistance to antimicrobial peptides. Indeed, we demonstrate that *mgrB* mutation confers resistance to the *G. mellonella* antimicrobial peptides. Notably, to restore the virulence of the *mgrB* mutant to wild-type levels, it was necessary to mutate all loci responsible for the lipid A modifications (Fig 2E). However, since in a previous work, we demonstrated that these individual mutants are already attenuated in *G. mellonella,* it is possible that there are additional, as yet unknown, MgrB-controlled factor(s) playing a role in *Galleria* virulence (Insua *et al*, 2013). At present, we can only speculate which are these factors. Recent transcriptional profiling analysis of six *mgrB* mutants revealed the upregulation of the several loci of the PhoPQ regulon but also the increased expression of two additional two-component regulatory systems, RstAB and CrrAB (Wright *et al*, 2015). Whereas the RstB sensor has been shown to control the expression of PhoP-regulated genes, there is no data on the CrrAB regulon and whether this system contributes to *K. pneumoniae* virulence. Future research efforts will be directed to understand the role of these systems in *Klebsiella* pathogenesis.

We demonstrated no impact of the *mgrB* mutation on the capacity of *K. pneumoniae* to establish pulmonary and systemic infection in the pneumonia mouse model. This finding is in contrast to the hypervirulence phenotype observed in *G. mellonella*, but this most likely reflects the complexity of mammalian models showing a more complex innate immune response dependent on several cell types. Nevertheless, in the setting of an immunocompromised host (e.g. ICU patient) coupled with increased AMP resistance, it is feasible that *mgrB* inactivation may play a role in the early establishment of

infection, pathogenicity and subsequent patient outcomes. Supporting this notion, isolation of colistin-resistant strains has been associated with death in septic patients with *K. pneumoniae* infection (Falcone *et al*, 2016).

Another novel finding of this work is that the *mgrB* mutant elicited limited activation of inflammatory responses *in vitro* and *in vivo* (*G. mellonella* model). This is particularly relevant since more than two decades of research have established the critical importance of a controlled inflammatory response to clear *K. pneumoniae* infection. Therefore, the reduced response induced by the *mgrB* mutant could be considered another feature of increased virulence associated with this mutation. Strikingly, *mgrB*-dependent attenuation of *G. mellonella* defence responses was solely dependent on LpxO-mediated lipid A modification. This is in perfect agreement with our recent work demonstrating that *Klebsiella* LPS containing 2-hydroxymyristate is less inflammatory than that lacking this modification (Llobet *et al*, 2015). In contrast, our data revealed that *mgrB*-mediated attenuation of inflammatory responses in macrophages was dependent on all four lipid A modifications. This reflects the complex interaction between *K. pneumoniae* and macrophages and is in good agreement with published evidence demonstrating the non-redundant contribution of *Klebsiella* lipid A modifications to limit phagocytosis by professional phagocytes (macrophages and amoeba; March *et al*, 2013). MAPKs play a crucial role in governing immune responses, thereby highlighting the relevance of attenuating their activation by the *mgrB* mutant (Dong *et al*, 2002). Current evidence shows that JNK and ERK regulate the induction of inflammatory responses and production of defensins, two crucial host defence responses against *K. pneumoniae* (Moranta *et al*, 2010), and in this work, we have shown that inactivation of *mgrB* resulted in limited activation of both MAPKs. Nonetheless, there is still limited understanding of which MAPKs-governed responses are crucial against *K. pneumoniae* infections. Future studies will be directed to identify these responses.

To explain how *mgrB*-controlled lipid A limits the activation of inflammatory responses, it is reasonable to postulate that this extensively modified lipid A is not recognised by the TLR4/MD-2 LPS receptor complex. The work of Park and co-workers predicts that the hepta-acylated lipid A containing palmitate will not be accommodated within the active site of its MD-2 receptor, hence blocking the subsequent dimerisation of TLR4 (Park *et al*, 2009). In addition, the substitution of the lipid A phosphates with Ara4N or PEtN should affect the receptor–ligand interaction since the lipid A 1- and 4′-phosphate groups interact with a cluster of positively charged residues from TLR4 and MD-2 (Park *et al*, 2009). To rigorously validate these predictions, it will be necessary to assess individually all these lipid A species with and without their various modifications. The elegant synthetic biology approach described by Stephen Trent's group (Needham *et al*, 2013) engineering of *E. coli* to produce each of the lipid A variants may provide a means for comparing the different lipid As.

In general, it is accepted that antibiotic resistance most often confers a cost in terms of subdued bacterial fitness and virulence (Beceiro *et al*, 2013). Previous studies also illustrate that microbial immune-evasive strategies contribute to the development and persistence of antimicrobial resistance (Needham & Trent, 2013; De Majumdar *et al*, 2015). Here, we have demonstrated a divergent

                                        

scenario whereby an antimicrobial resistance mechanism helps to counteract the activation of immune defences and enhance virulence. Overall, this evidence stresses the importance of considering antimicrobial resistance and virulence together, while also highlighting the urgent need to include the identification of virulent clones in clinical microbiology laboratories.

Finally, it is worthwhile discussing the clinical consequences and implications of our findings. Limited therapeutic options to treat MDR Gram-negative pathogens, in particular *Pseudomonas aeruginosa*, *Acinetobacter baumannii* and *K. pneumoniae*, led clinicians to reappraise the clinical application of colistin. This antibiotic is increasingly being used in endemic areas for KPC-producing *K. pneumoniae*, but reports of colistin-resistant isolates in that species are on the rise (Tzouvelekis *et al*, 2012; Olaitan *et al*, 2014b; Nation *et al*, 2015). The evidence presented in this work indicates that inactivation of *mgrB* results in colistin resistance, but also enhances *K. pneumoniae* virulence. The latter finding is particularly alarming because a large number of *Klebsiella* strains resistant to colistin carry alterations of the *mgrB* gene (Lippa & Goulian, 2009; Cannatelli *et al*, 2013; Olaitan *et al*, 2014a; Poirel *et al*, 2015; Wright *et al*, 2015; Zowawi *et al*, 2015). In fact, the heightened virulence of these strains might be one of the explanations underlying the increased mortality associated with these infections (Capone *et al*, 2013). Physicians should be aware of such an occurrence for its implications on treatment and outcome. This evidence further stresses the importance of careful consideration of colistin therapies, at least for *Klebsiella* infections. In this context, we put forward the need to perform well-designed clinical trials to inform antibiotic regimens for treating infections caused by KPC-*Klebsiella*.

# Materials and Methods

### Ethics statement

The experiments involving mice were approved by the Queen's University Belfast's Ethics Committee and conducted in accordance with the UK Home Office regulations (Project Licence PPL2700). Female C57BL/6 mice (8–9 weeks of age) were mock-infected with PBS ($n = 4$) and infected with the wild-type strain ($n = 6$) or the *mgrB* mutant ($n = 6$). Animals were randomised for interventions, but researchers processing the samples and analysing the data were aware which intervention group corresponded to which cohort of animals.

### Bacterial strains and growth conditions

Bacterial strains and plasmids used in this study are presented in Appendix Table S3. Kp52145 is a clinical isolate, serotype O1:K2, belonging to the virulent CC65 clonal complex (Brisse *et al*, 2009; Lery *et al*, 2014). Six previously published clinical *K. pneumoniae* strains (T1a, T1b, C21, C22, C2 and 15I5) were supplied by Prof. Patrice Nordmann, University of Fribourg, Switzerland (Poirel *et al*, 2015). Bacteria were grown in LB medium at 37°C, and where appropriate, antibiotics were supplemented at the following concentrations: ampicillin (Amp) 100 μg/ml, trimethoprim (Tmp) 100 μg/ml, tetracycline (Tet) 12.5 μg/ml, chloramphenicol (Cm) 25 μg/ml, kanamycin (Km) 50 μg/ml and carbenicillin (Cb) 50 μg/ml.

### Mutagenesis of *K. pneumoniae* 52145

PCR primers used for the *mgrB* mutant construction were designed using the whole genome sequence of Kp52145 (GenBank Accession No. FO834906.1; Appendix Table S3). The primer pairs *mgrB*_UPFWD, *mgrB*_UPRVS, *mgrB*_DWNFWD and *mgrB*_DWNRVS (Appendix Table S4) were used in separate PCR reactions to amplify 550- to 800-bp fragments flanking the *mgrB* gene. BamHI restriction sites internal to these flanking regions were incorporated at the end of each amplicon. Purified *mgrB* UP and DOWN fragments were then polymerised and amplified as a single PCR amplicon using the primers *mgrB*_UPFWD and *mgrB*_DWNRVS. This 1.4-kb PCR amplicon was then cloned into pGEM-T Easy (Promega) to obtain pGEMΔ*mgrB* and transformed into *E. coli* C600. After EcoRI digestion, the purified 1.4-kb fragment was cloned into EcoRI-digested Antarctic Phosphatase (New England Biolabs)-treated pGPI-SceI-2 suicide vector (Aubert *et al*, 2014) to generate pGPI-SceI-2Δ*mgrB* and transformed into *E. coli* GT115 (or SY327). pGPI-SceI-2Δ*mgrB* was thereafter transformed into the diaminopimelate (DAP) auxotrophic *E. coli* donor strain β2163 (Demarre *et al*, 2005) and mobilised into *K. pneumoniae* via conjugation. Selection of co-integrant clones was undertaken using LB agar supplemented with Tmp at 37°C. A second crossover reaction was then performed by conjugating the pDAI-SceI-SacB plasmid (Aubert *et al*, 2014) into a refreshed overnight culture containing three Tmp-resistant co-integrant clones. Exconjugants were selected on LB agar supplemented with Tet at 37°C. Candidate mutant clones were checked for susceptibility to Tmp and then confirmed by PCR using the *mgrB*_UPFWD and *mgrB*_DWNRVS primers. Curing of the pDAI-SceI-SacB vector was performed by plating a refreshed overnight culture of one *K. pneumoniae* mutant colony onto 6% sucrose LB agar without NaCl at 30°C for 24 h. A single clone surviving the sucrose treatment was checked for susceptibility to Tet, confirmed by PCR and named 52145-Δ*mgrB*.

The 52145-Δ*pmrC* and 52145-Δ*pagP* mutants were constructed by inserting a linearised pKD4-derived Km resistance cassette (primers: *pmrC*_FWD, *pmrC*_RVS, *pagP*_FWD and *pagP*_RVS, accordingly; Appendix Table S4) into the relevant gene using the lambda Red recombinase method and pKOBEG-sacB plasmid (Datsenko & Wanner, 2000; Derbise *et al*, 2003). Km cassette removal was undertaken by Flp-mediated recombination using the pFLP2Tp plasmid (Hoang *et al*, 1998). The 52145-Δ*pmrC* and 52145-Δ*pagP* strains were confirmed by PCR using the primers *pmrC*_checkFWD, *pmrC*_checkRVS, *pagP*_checkFWD and *pagP*_checkRVS, accordingly (Appendix Table S4).

The double-mutant 52145-Δ*mgrB*-Δ*phoQGB* was obtained by conjugating pMAKSACΔ*phoPQ*GB into 52145-Δ*mgrB* as previously described (Llobet *et al*, 2011). Double 52145-Δ*pmrC*-Δ*mgrB* and 52145-Δ*pagP*-Δ*mgrB* mutants were generated by conjugating pGPI-SceIΔ*mgrB* into 52145-Δ*pmrC* and 52145-Δ*pagP* (Llobet *et al*, 2015), respectively. The double 52145-Δ*mgrB*-Δ*lpxO* mutant was created by conjugating pMAKSACΔ*lpxO* into 52145-Δ*mgrB*. The pMAKSACΔ*lpxO* vector was constructed in accordance with the previously described mutagenesis strategy using DNA fragments amplified by the primers *lpxO*_UPFWD, *lpxO*_UPRVS, *lpxO*_DWNFWD and *lpxO*_DWNRVS (Appendix Table S4; Llobet *et al*, 2011). The triple 52145-Δ*pmrC*-Δ*lpxO*-Δ*mgrB* mutant was generated by conjugating the pMAKSACΔ*lpxO* construct into 52145-Δ*pmrC* and then

conjugating the pGPI-SceIΔ*mgrB* into the 52145-Δ*pmrC*-Δ*lpxO* double mutant.

52145-Δ*mgrB*-Δ*pmrAB*, 52145-Δ*mgrB*-Δ*pmrF*, 52145-Δ*mgrB*-Δ*lpxO*-Δ*pmrF* and 52145-Δ*pmrC*-Δ*lpxO*-Δ*mgrB*-Δ*pmrF*, 52145-Δ*pmrC*-Δ*lpxO*-Δ*mgrB*-Δ*pmrF*-Δ*pagP*, 52145-Δ*manC* and 52145-Δ*mgrB*-Δ*manC* mutants were constructed using the pGPI-SceI-2 and pDAI-SceI-SacB plasmid methodology detailed for the *mgrB* mutant. PCR fragments flanking the *pmrAB*, *pmrF*, *pagP* and *manC* genes were initially generated using the relevant UPFWD, UPRVS, DWNFWD and DWNRVS primers (Appendix Table S4). Gene-specific UPFWD and DWNRVS primers were used to generate Δ*pmrAB*, Δ*pmrF*, Δ*pagP* and Δ*manC* knockout fragments and confirm mutant clones. pGPI-SceI-2Δ*pmrAB*, pGPI-SceI-2Δ*pmrF*, pGPI-SceI-2Δ*pagP* and pGPI-SceI-2Δ*manC* constructs were then conjugated with the relevant 52145-wild-type and mutant precursor strains. Confirmation of all mutants was undertaken by PCR.

## Complementation of the 52145-Δ*mgrB* mutant

For complementation of the 52145-Δ*mgrB* mutant strain, a PCR fragment (primers: *mgrB*_UPFWD and *mgrB*_DWNRVS) comprising the coding and promoter regions of the *K. pneumoniae* 52145 *mgrB* gene was amplified using Phusion® High-Fidelity DNA Polymerase (New England Biolabs). The 1,507-bp amplicon was gel-purified and then cloned into SmaI-digested (New England Biolabs), Antarctic Phosphatase (New England Biolabs)-treated pUC18R6KT-mini-Tn7TKm plasmid (Choi *et al*, 2005) to obtain pUC18R6KT-mini-Tn7TKm_Kp52145*mgrB*Com. pUC18R6KT-mini-Tn7TKm_Kp52145 *mgrB*Com was then transformed into *E. coli* SY327 and thereafter into *E. coli* β2163. In addition, the transposase-containing pTSNSK-Tp plasmid (Crepin *et al*, 2012) was introduced to the 52145-Δ*mgrB* strain by electroporation to give 52145-Δ*mgrB*/pTSNSK-Tp. 52145-Δ*mgrB*/pTSNSK-Tp was then conjugated overnight with *E. coli* β2163/pUC18R6KT-mini-Tn7TKm_Kp52145*mgrB*Com on LB agar supplemented with DAP at 30°C. The retrieved culture was then serially diluted in sterile PBS, plated onto LB Km agar and incubated at 42°C for 6.5 h followed by 37°C overnight. The colonies grown were thereafter screened for resistance to Km and susceptibility to Tmp and Amp. Correct integration of the Tn7 transposon was confirmed by PCR using the primers KpnglmSup/Ptn7L and KpnglmSdown/Ptn7R as previously described (March *et al*, 2013). Additionally, the presence of the *mgrB* gene was PCR-confirmed using the *mgrB*_UPFWD and *mgrB*_DWNRVS primers.

## Complementation of the 52145-Δ*mgrB*-Δ*phoQ*GB and 52145-Δ*mgrB*-Δ*lpxO* mutants

Complementation of the 52145-Δ*mgrB*-Δ*phoQ*GB mutant with the 52145 *phoPQ* operon and 52145-Δ*mgrB*-Δ*lpxO* mutant with the 52145 *lpxO* gene was undertaken using pGP-Tn7-Cm_KpnPhoPQCom and pGP-Tn7-Cm_KpnLpxOCom as previously described (Llobet *et al*, 2015). Complemented strains were identified by resistance to Cm, susceptibility to Tmp and Amp, and PCR using the primers PhoPQ_ checkFWD, PhoPQ_checkRVS, *lpxO*_checkFWD and *lpxO*_ checkRVS, accordingly (Appendix Table S4). Correct Tn7 integration was PCR-confirmed using the KpnglmSup/Ptn7L and KpnglmSdown/ Ptn7R primer sets (March *et al*, 2013).

## Growth curve analysis

For growth analyses, 5 μl of overnight cultures was diluted in 250 μl of LB or M9 minimal medium (5× M9 minimal salts [Sigma-Aldrich] supplemented with 2% glucose, 3 mM thiamine, 2 mM MgSO₄) and incubated at 37°C with continuous, normal shaking in a Bioscreen C™ Automated Microbial Growth Analyzer (MTX Lab Systems, Vienna, VA, USA). Optical density (OD; 600 nm) was measured and recorded every 20 min.

## Biofilm analysis

Biofilms were assayed using a variation of the standard crystal-violet quantification assay. Overnight cultures were grown for 16 h in 5 ml LB broth, after which each strain was diluted to an $OD_{600}$ of 0.02 in freshly prepared M9 minimal media (5× M9 minimal salts [Sigma-Aldrich] supplemented with 2% glucose, 3 mM thiamine, 2 mM MgSO₄). 100 μl of each diluted strain was then added to two columns (12 internal wells) of a polystyrene U-bottomed 96-well plate (Greiner Bio-One), and the plate was incubated statically at 37°C for 24 h. Wells were stained with the addition of 25 μl of 0.5% crystal violet (Sigma-Aldrich) to each well for 1 h. The plates were washed by submersion in distilled H₂O, and the crystal violet in each well was dissolved in 150 μl 95% ethanol for 1 h. Biofilms were then quantified by measuring the optical density at 595 nm of each well. Three independent cultures of each strain were tested per day, and the data shown are the average of three independent days.

## Polymyxin and antimicrobial peptide susceptibility assays

MICs to colistin sulphate and polymyxin B were determined by Etest® (bioMérieux) using the European Committee on Antimicrobial Susceptibility Testing (EUCAST) breakpoints for colistin (the European Committee on Antimicrobial Susceptibility Testing, 2016).

To assay *K. pneumoniae* strains for resistance to polymyxin antibiotics and AMPs, we used a modified version of the sensitivity assay described by Llobet *et al* (2011). Briefly, each strain was grown to early exponential phase in LB broth, washed once in PBS and diluted in liquid testing media (1% v/v Tryptone soy broth, 10% v/v 100 mM phosphate buffer [pH 6.5], 2% v/v 5 M NaCl) to an approximate concentration of $4 \times 10^4$ colony-forming units (CFUs) per millilitre. Twenty-five microlitres of each diluted strain was then mixed with 5 μl of antibiotic (or AMP) and incubated at 37°C for 1 h. Fifteen microlitres of the suspension was thereafter spread onto LB agar and incubated overnight at 37°C. Per cent survival of the cells exposed to the antibiotics (or AMPs) was determined through comparison with the unexposed (sterile PBS) controls.

## Lipid A isolation and mass spectrometry

Lipid A was extracted using the ammonium hydroxide/isobutyric acid method described earlier (El Hamidi *et al*, 2005). Negative ion MALDI-TOF mass spectrometry analysis (Bruker Daltonics) of the samples was undertaken using an equal volume of dihydroxybenzoic acid matrix (Sigma-Aldrich) dissolved in (1:2) acetonitrile-0.1% trifluoroacetic acid.

### Generation of the *lucFF* reporter fusion *K. pneumoniae* strains

An 868-bp amplicon comprising the *pagP* gene promoter region was amplified by Phusion® High-Fidelity DNA Polymerase using the primers *pagP*_Pro_FWD and *pagP*_Pro_RVS (Appendix Table S4). The amplicon was digested with EcoRI, gel-purified, cloned into an EcoRI-SmaI-digested pGPL01 suicide vector and then transformed into *E. coli* GT115 cells to obtain pGPLKpnProPagP. Correct insertion of the amplicon was verified by restriction digestions with EcoRI and HindIII.

pGPLKpnProPhoP, pGPLKpnProLpxO, pGPLKpnProPmrH, pGPLKpnProPmrC, pGPLKpnProPagP and pGPLKpnProcps (Llobet *et al*, 2011, 2015; Insua *et al*, 2013) were each introduced into *E. coli* β2163 and then mobilised into the Kp52145, 52145-Δ*mgrB*, 52145-Δ*mgrB*Com and 52145-Δ*mgrB*-Δ*phoQ*GB strains via conjugation. Cultures were then serially diluted and checked for Amp resistance by plating on LB Cb agar at 37°C. Correct insertion of the vectors into the chromosome was confirmed by PCR using the relevant *lucFF*_check and promoter sequence primers (Appendix Table S4; data not shown).

### Luciferase activity

Overnight cultures of the *K. pneumoniae* reporter strains were refreshed for 2.5 h in LB containing Cb at 37°C and 180 rpm. The cells were then pelleted, washed once in sterile PBS and adjusted to an $OD_{600}$ of 1.0. One hundred microlitres of each suspension was added to an equal volume of luciferase assay reagent (1 mM D-luciferin [Synchem] in 100 mM sodium citrate buffer pH 5.0), vortexed for 5 s and then immediately measured for luminescence (expressed as relative light units [RLU]) using a GloMax 20/20 Luminometer (Promega). All strains were tested in triplicate from three independent cultures.

### *G. mellonella* larvae and infections

*G. mellonella* larvae were obtained from UK Waxworms Limited. Upon receipt, larvae were stored in reduced light at 13°C with nil dietary supplementation. All experiments were performed within 14 days of receipt comprising larvae showing a healthy external appearance of 250–350 mg weight as previously described (Insua *et al*, 2013).

*K. pneumoniae* strains for *G. mellonella* infections were prepared by harvesting refreshed 5 ml of exponential phase LB cultures (37°C, 180 rpm, 2.5 h), washing once in sterile PBS and then adjusting to an $OD_{600}$ of 1.0 (i.e. ~$5 \times 10^8$ CFUs/ml). Each suspension was thereafter diluted to the desired working concentration (i.e. ~$1 \times 10^7$ and $1 \times 10^8$ CFUs/ml). Larvae were surface-disinfected with 70% (v/v) ethanol and then injected with 10 μl of working bacterial suspension at the right last proleg using a Hamilton syringe equipped with a 27-gauge needle. Experiments involving dual injections were undertaken in the right and then the left proleg. For each experiment, 10 larvae injected with sterile PBS were included as combined trauma and vehicle controls. Injected larvae were placed inside Petri dishes at 37°C in the dark. Per cent survival following gentle physical stimulation was recorded at 24-h intervals over 72 h.

### *G. mellonella* killing assay

Based on earlier 50% lethal dose ($LD_{50}$) data, the virulence of each *K. pneumoniae* 52145 strain was explored using a working concentration of ~$1 \times 10^5$ CFUs per larvae (Insua *et al*, 2013). For *G. mellonella* infections using the *K. pneumoniae* T1a and T1b ST-258 clinical strains, an infection dose of ~$2 \times 10^5$ per larvae was used (Poirel *et al*, 2015). Insects were considered dead if they did not respond to physical stimuli. Larvae were examined for pigmentation, and time of death was recorded. Assays were allowed to proceed for only 3 days as pupa formation was occasionally observed by day 4. A total of 40 larvae over three independent experiments were investigated for each strain.

### *G. mellonella* with boosted immunity killing assay

Larvae were injected with ~$1 \times 10^6$ CFUs of heat-killed (65°C, 20 min) *E. coli* MG1655 to elicit the production of antimicrobial factors (Insua *et al*, 2013). After 24 h of incubation at 37°C, larvae were then infected with $1 \times 10^6$ CFUs of selected *K. pneumoniae* 52145 strains ($n = 30$ larvae/group over three independent experiments) and returned to 37°C for 72 h. Per cent survival was recorded at 24-h intervals.

### Radial diffusion bioassay to assess *G. mellonella* antimicrobial peptides

We used a previously described radial diffusion assay to assess the *in vitro* resistance of the *K. pneumoniae* 52145 strains to antimicrobial factors produced by *G. mellonella* (Insua *et al*, 2013). Briefly, production of *G. mellonella* antimicrobial factors was elicited through injection of ~$1 \times 10^6$ CFUs of heat-killed *E. coli* MG1655. After 24 h of incubation at 37°C, 15 μl of haemolymph from three surface-disinfected larvae was pooled with 10 μl of saturated *N*-phenylthiourea (Sigma-Aldrich). Ten microlitres of each haemolymph preparation (or sterile $H_2O$) was then inoculated into the wells of radial diffusion underlay gels containing either 52145, 52145-Δ*mgrB* or 52145-Δ*mgrB*Com. Plates were incubated at 37°C for 3 h, overlayed with 1% agarose containing 6% TSB powder and returned to 37°C for 18 h. Zones of inhibition were measured and expressed as inhibition units (10 units = 1 mm). Haemolymph samples from larvae injected with sterile PBS were used as uninfected controls. Colistin sulphate and polymyxin B (20 μg/ml) plus sterile $H_2O$ were used as positive and negative bioassay controls, respectively.

### *G. mellonella* RNA extraction and quantitative real-time PCR analysis

Larvae were infected with ~$1 \times 10^5$ CFUs of the selected *K. pneumoniae* 52145 strains, incubated at 37°C for 8 h and then homogenised in 1 ml ice-cold TRIzol (Ambion™) using a VDI 12 tissue homogeniser (VWR). Total RNA was extracted according the manufacturer's instructions with minor modifications including the use of Phase Lock Gel Heavy 2-ml tubes (VWR) for phase separation. Five micrograms of RNA was treated with recombinant DNase I (Roche Diagnostics Ltd) at 37°C for 30 min and then purified using a standard phenol–chloroform method and Phase Lock Gel Heavy 1.5-ml tubes (VWR). The RNA was precipitated overnight with 20 μl 3 M sodium

acetate (pH 5.2) and 600 μl 98% (v/v) ethanol at −20°C, washed twice in 75% (v/v) ethanol, dried and then resuspended in RNase-free $H_2O$. Duplicate cDNA preparations from each sample were generated from 1 μg of RNA using Moloney murine leukaemia virus (M-MLV) reverse transcriptase (Sigma-Aldrich) according to the manufacturer's instructions. Quantitative real-time PCR analysis of *G. mellonella* antimicrobial peptide expression was undertaken using the KAPA SYBR® FAST qPCR Kit, previously described oligonucleotide primers (Insua *et al*, 2013) and Stratagene Mx3005P qPCR System (Agilent Technologies). Thermal cycling conditions were as follows: 95°C for 3 min for enzyme activation, 40 cycles of denaturation at 95°C for 10 s and annealing at 60°C for 20 s. cDNA samples were tested in duplicate, and relative mRNA quantity was determined by the comparative threshold cycle ($\Delta\Delta C_t$) method using 18S rRNA normalisation.

## Capsule polysaccharide purification and quantification

Bacteria were grown overnight in 3 ml of LB medium (37°C, 180 rpm) with viable counts determined by dilution plating. The cultures were then harvested by centrifugation, and the cell pellet was resuspended in 500 μl of sterile $H_2O$. Each sample was then treated with 1% 3-(*N,N*-dimethyltetradecylammonio)propanesulphonate (Sigma-Aldrich; in 100 mM citric acid, pH 2.0) at 50°C for 20 min. Bacterial debris was pelleted ($3,220 \times g$, 10 min), and 250 μl of the supernatant was transferred to a clean 15-ml glass tube. The CPS was thereafter ethanol-precipitated at −20°C for 20 min and pelleted ($9,447 \times g$, 10 min, 4°C). After removal of the supernatant, the pellet was then dried (5 min, 90°C) and resuspended in 200 μl of sterile water. CPS quantification was undertaken by determining the concentration of uronic acid in the samples, using a modified carbazole assay as previously described (Rahn & Whitfield, 2003). All samples were tested in triplicate.

## Intranasal murine infection model

Female mice were infected intranasally with ~$3 \times 10^5$ Kp52145 and 52145-Δ*mgrB* in 30 μl PBS (*n* = 6 per strain). Control mice were inoculated with 30 μl sterile PBS (*n* = 4). After 24 h, mice were euthanised using a Schedule 1 method according to UK Home Office-approved protocols. Left lung samples from infected and uninfected control mice were immersed in 1 ml of RNA stabilisation solution (50% [w/v] ammonium sulphate [Fisher Scientific], 2.9% [v/v] 0.5 M ethylenediaminetetraacetic acid [Sigma-Aldrich], 1.8% [v/v] 1 M sodium citrate [Sigma-Aldrich]) on ice and then stored at 4°C for at least 24 h prior to RNA extraction. Right lung, spleen and NALT samples from infected mice were immersed in 1 ml sterile PBS on ice and processed for quantitative bacterial culture immediately. Samples were homogenised using a VDI 12 tissue homogeniser, serially diluted in sterile PBS and plated onto Salmonella Shigella agar (Oxoid Limited), and the colonies were enumerated after overnight incubation at 37°C. Data were expressed as CFUs per sample.

## RNA extraction and quantitative real-time PCR analysis of infected murine lung

Left lung samples for RNA extraction were removed from RNA stabilisation solution, homogenised in 1 ml ice-cold TRIzol

(Ambion™) using a VDI 12 tissue homogeniser and then subjected to five rounds of bead beating for 10 s followed by ice-cooling for 1 min using 100-μm acid-washed glass beads (Sigma-Aldrich) and a Mini-Beadbeater-1 (BioSpec Products). RNA was extracted, and duplicate cDNA preparations were generated from each sample using the protocol described for the infected *G. mellonella* samples. Quantitative real-time PCR analysis of inflammatory cytokine and murine antimicrobial peptide expression was undertaken using the KAPA SYBR® FAST qPCR Kit, previously described oligonucleotide primers (Appendix Table S4) and Stratagene Mx3005P qPCR System (Agilent Technologies). Thermal cycling conditions were as follows: 95°C for 3 min for enzyme activation, 40 cycles of denaturation at 95°C for 10 s and annealing at 60°C for 20 s. Each cDNA sample was tested in duplicate, and relative mRNA quantity was determined by the comparative threshold cycle ($\Delta\Delta C_t$) method using hypoxanthine phosphoribosyltransferase 1 (*mHprt*) gene normalisation.

## Cell culture and infections

Immortalised BMDM (iBMDM) cells (BEI Resources, NIAID, NIH: Macrophage Cell Line Derived from Wild Type Mice, NR-9456) were grown in Dulbecco's modified Eagle's medium (DMEM; Gibco® 41965) supplemented with 10% heat-inactivated foetal calf serum (FCS), 100 U/ml penicillin and 0.1 mg/ml streptomycin (Gibco) at 37°C in a humidified 5% $CO_2$ incubator. Murine alveolar macrophages MH-S (ATCC, CRL-2019) were grown in RPMI 1640 tissue culture medium supplemented with 10% heat-inactivated foetal calf serum (FCS), 100 U/ml penicillin, 0.1 mg/ml streptomycin (Gibco), and 10 mM HEPES (Sigma-Aldrich). iBMDMs and MH-S were grown at 37°C in humidified 5% $CO_2$ atmospheric conditions. Cells were routinely tested for *Mycoplasma* contamination.

For infections, bacteria were adjusted to an $OD_{600}$ of 1.0 in PBS, and infections were performed using a multiplicity of infection of 100 bacteria per cell. To synchronise infection, plates were centrifuged at $200 \times g$ for 5 min.

## Immunoblot analysis

Proteins were resolved by standard 10% SDS–PAGE and electroblotted onto nitrocellulose membranes. Membranes were blocked with 4% bovine serum albumin (w/v) in TBST, and protein bands were detected with specific antibodies using chemiluminescence reagents and a G:BOX Chemi XRQ chemiluminescence imager (Syngene).

The following rabbit antibodies were used: anti-IκBα (1:1,000; Cell Signaling), anti-phospho-p38 (1:1,000; Cell Signaling), anti-phospho-SAPK/JNK (1:1,000; Cell Signaling) and anti-phospho-ERK (1:1,000; Santa Cruz Biotechnology). Immunoreactive bands were visualised by incubation with horseradish peroxidase-conjugated goat anti-rabbit immunoglobulins (1:5,000) or goat anti-mouse immunoglobulins (1:1,000; Bio-Rad). To ensure that equal amounts of proteins were loaded, blots were re-probed with α-tubulin (1:3,000; Sigma-Aldrich).

To detect multiple proteins, membranes were re-probed after stripping of previously used antibodies using a pH 2.2 glycine-HCl/SDS buffer.

## Quantification of cytokines

Infections were performed in 96-well plates ($5 \times 10^4$ cells per well) using $1 \times 10^5$ UV-killed bacteria (1 ml of bacterial suspension adjusted to an $OD_{600}$ of 1.0 was subjected to 10 J UV light for 30 min, and bacterial killing was confirmed by plating in LB). TNF-$\alpha$ in the supernatants was determined at 6 h of infection using a Murine TNF-$\alpha$ Standard TMB ELISA Development Kit (PeproTech, catalogue number 900-T54), according to the manufacturer's instructions. Experiments were performed in duplicate and repeated at least three times.

## Statistical analyses

Statistical analyses were performed using the two-tailed *t*-test, or when the requirements were not met, by the Mann–Whitney *U*-test. *P*-values of < 0.05 were considered statistically significant. Normality and equal variance assumptions were tested with the Kolmogorov–Smirnov test and the Brown–Forsythe test, respectively. Survival analyses were undertaken using the log-rank (Mantel–Cox) test with Bonferroni correction for multiple comparisons ($\alpha$ = 0.008). All analyses were performed using GraphPad Prism for Windows (version 5.03) software.

**Expanded View** for this article is available online.

## Acknowledgements

We thank Professor Patrice Nordmann (University of Fribourg, Switzerland) for the provision of the clinical *K. pneumoniae* isolates, as well as the members of the JAB laboratory for their thoughtful discussions and support with this project. We also gratefully acknowledge Mr. Ian Brennan (Trinity College Dublin, Ireland) and Ms. Lydia Roets (Queen's University Belfast, United Kingdom) for their assistance with creating some of the mutant strains, as well as Dr. Verónica Martínez Moliner (University of the Balearic Islands, Spain) and Ms. Barbara Corelli (Pasteur Institute, France) for the design and creation of the pMAKSACBΔlpxO and pGPI-SceI-2ΔmanC constructs used herein. GM is the recipient of a PhD Fellowship funded by the Department for Employment and Learning (Northern Ireland, UK). LH is the recipient of a Queen's University Belfast Research Fellowship. TJK is the recipient of an ERS-EU RESPIRE2 Marie Skłodowska-Curie Postdoctoral Research Fellowship—MC RESPIRE2 1st round 4571-2013 and a National Health and Medical Research Council Early Career Fellowship (GNT1088448). The research leading to these results has received funding from the People Programme of the European Union's Seventh Framework Programme (FP7/2007-2013) under REA grant agreement 600368. This work was supported by Marie Curie Career Integration Grant U-KARE (PCIG13-GA-2013-618162); Biotechnology and Biological Sciences Research Council (BBSRC, BB/L007223/1, BB/N00700X/1 and BB/P006078/1); and Queen's University Belfast start-up funds to JAB.

## Author contributions

TJK and JAB conceived the study and wrote the first draft of the manuscript. TJK, GM, JS-P, CGF, AD, JLI and LH performed the experiments and contributed data for this work. TJK, GM, JS-P, CGF, AD, JLI, RI, LH and JAB contributed to and approved the final version of the manuscript.

## Conflict of interest

The authors declare that they have no conflict of interest.

## The paper explained

### Problem

The emergence of multidrug-resistant (MDR) *K. pneumoniae* is an important public health challenge worldwide. Of particular concern is the recent appearance of MDR *K. pneumoniae* strains that develop colistin resistance following treatment with this last-line antibiotic agent. Several studies have shown that the development of colistin resistance is often associated with mutational inactivation of the *mgrB* gene. However, the precise mechanisms governing the colistin resistance observed in these organisms are poorly understood. Moreover, it is currently unknown whether *mgrB* mutation confers any loss of virulence. This is particularly critical given the increasing number of *K. pneumoniae* infections caused by virulent clones and the ease with which *mgrB* mutations arise in the hospital setting.

### Results

Here, we constructed and characterised an *mgrB* mutant in a clinically relevant *K. pneumoniae* strain. Our data show that *mgrB* inactivation results in bacterial outer membrane modifications, which not only confers resistance to colistin, but also to host defence peptides produced by the body to help combat infection. Surprisingly, *mgrB* mutation substantially increased *K. pneumoniae* virulence in an established invertebrate infection model, and its survival was not compromised in the mouse. Our data also indicate that this virulence phenotype may be linked to subdued host immune system activation during the early infection.

### Impact

Our findings have important implications for the management of patients with MDR *K. pneumoniae*. In the clinical settings such as intensive care units, it is feasible that *mgrB* inactivation may play a role in the early infection establishment, pathogenicity and patient outcomes. This reinforces the importance of considering antimicrobial resistance and virulence together, while also highlighting the importance of microbiological surveillance for virulent clones in healthcare settings. Further, our findings stress that care is required in the use of colistin against infections caused by *K. pneumoniae*.

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
