## [Review Process File · EMBO Molecular Medicine]

***A Klebsiella pneumoniae* antibiotic resistance mechanism that subdues host defences and promotes virulence**

Timothy Kidd, Grant Mills, Joana Sá-Pessoa, Amy Dumigan, Christian Frank, José Insua, Rebecca Ingram, Laura Hobley, and José Bengoechea

Corresponding author: José Bengoechea, Queen's University Belfast

Review timeline:

Submission date:	14 November 2016
Editorial Decision:	15 December 2016
Revision received:	22 December 2016
Editorial Decision:	09 January 2017
Revision received:	11 January 2017
Accepted:	12 January 2017

Transaction Report:

Editor: Céline Carret

1st Editorial Decision

15 December 2016

Thank you for the submission of your manuscript to EMBO Molecular Medicine. We have now heard back from the two referees whom we asked to evaluate your manuscript. Although the referees find the study to be of potential interest, they also raise a few concerns that need to be addressed in the next final version of your article.

You will see from the comments below that both referees are enthusiastic about the work and only minor issues are raised. Notwithstanding, for our scope and interest, the manuscript does not provide any direct clinical implications at this stage. I would like to encourage you to thoroughly discuss the clinical consequences and implications of your findings to increase their medical relevance.

We would welcome the submission of a revised version for further consideration and depending on the nature of the revisions, this may be sent back to the referees for another round of review.

Please note that it is EMBO Molecular Medicine policy to allow only a single round of revision and that, as acceptance or rejection of the manuscript may depend on another round of review, your responses should be as complete as possible.

I look forward to receiving your revised manuscript.

***** Reviewer's comments *****

Referee #1 (Comments on Novelty/Model System):

the medical impact is definitely medium if not low; there is no immediate consequence. This is a basic research study.

Referee #1 (Remarks):

The multidrug resistance of *Klebsiella* is a great problem worldwide. In particular the authors aimed to elucidate what is the factor which develop colistin resistance in *Klebsiella*. Previous studies demonstrated that this was due to a mutational inactivation of *mgrB* gene. Still the authors wanted to clarify this at molecular level.

Through a series of studies, they reach the point that lipopolysaccharide lipid A is the chemical factor which confers resistance to the colistin by adding new chemical moieties to the pre-existing lipid A. It seems that in particular the presence of positively charged groups present in aminoarabinose and 2-amino ethanol work withdrawing the cation peptides, colistin included. This is a known mechanism; the presence of a further fatty acid, the 2-hydroxy-myristic was also shown. But for this no explanation/ speculation is given at molecular level. Additionally and intriguingly, the authors demonstrate that these chemical addition to the lipid A have the effect to render *Klebsiella* lipid A a weak stimulator of innate immunity. This was demonstrated *in vitro* and *in vivo* by a plethora of immunological experiments. So these mutations at chemical level switch the *Klebsiella* strain to be colistin resistant and also more virulent. This is certainly a novel finding, since it is thought that a higher resistance is paid by the bug with a low virulence.

I trust these are novel and interesting findings and the work is really good and experiments are well-conceived and carried out. The only part which is missing is at molecular level, there is no clear-cut explanation of the action/contribute (if any) of 2-hydroxy miristate to the resistance to polymyxins, only a vague statement; and there is no hypothesis of how this new lipid A could bind to TLR4/MD2 binary system in a different fashion, here maybe a deeper look at literature is worth. Maybe a couple of statements and a hypothesis could be advanced.

Referee #2 (Comments on Novelty/Model System):

This study is focused on the mechanism of antimicrobial peptide resistance in *K. pneumoniae*. Specifically, mutants deficient in the function of *mdrB* gene are analyzed. The authors used a comprehensive approach to link modifications in lipid A to regulatory and functional genes in *K. pneumoniae* and to the host responses. The technical quality is exceptional. Novelty is in the comprehensive approach, which takes previous speculations onto scientifically valid levels.

Referee #2 (Remarks):

An exceptional quality study that establishes functional and mechanistic links between mutational modifications of lipid A to specific genes and lipid A structures in *K. pneumoniae* and in the model hosts. The only minor suggestion is to rephrase the statement regarding the "conventional wisdom" or "dogma" that antibiotic resistance is linked to reduced bacterial fitness. This wisdom has been challenged in many publications in the last five years and by the simple fact of rapid spread in antibiotic resistance in clinics.

1st Revision - authors' response

22 December 2016

We appreciate the referee's efforts to assess our manuscript. We are glad that both of them share our enthusiasm about this work. Both referees raised two minor issues which we have met as follows (answer right after the referee's comment which is blue marked):

Referee 1:

The multidrug resistance of Klebsiella is a great problem worldwide. In particular the authors aimed to elucidate what is the factor which develop colistin resistance in Klebsiella Previous studies demonstrated that this was due to a mutational inactivation of mgrB gene. Still the authors wanted to clarify this at molecular level. Through a series of studies, they reach the point that lipopolysaccharide lipid A is the chemical factor which confers resistance to the colistin by adding new chemical moieties to the pre-existing lipid A. It seems that in particular the presence of positively charged groups present in aminoarabinose and 2-amino ethanol work withdrawing the cation peptides, colistin included. This is a known mechanism; the presence of a further fatty acid, the 2-hydroxy-myristic was also shown. But for this no explanation/ speculation is given at molecular level. Additionally and intriguingly, the authors demonstrate that these chemical addition to the lipid A have the effect to render Klebsiella lipid A a weak stimulator of innate immunity. This was demonstrated in vitro and in vivo by a plethora of immunological experiments. So these mutations at chemical level switch the Klebsiella strain to be colistin resistant and also more virulent. This is certainly a novel finding, since it is thought that a higher resistance is paid by the bug with a low virulence.

I trust these are novel and interesting findings and the work is really good and experiments are well-conceived and carried out.

We thank the referee for her/his very positive assessment of our study. S/he has nicely summarized the main findings our work while putting them in the context of the state-of-the-art.

The only part which is missing is at molecular level, there is no clear-cut explanation of the action/contribute (if any) of 2-hydroxy miristate to the resistance to polymyxins, only a vague statement;

Our findings (this work and our recent publication Llobet et al Proc Natl Acad Sci U S A. 2015 112(46):E6369-78) sustain the important role of 2-hydroxy myristate modification on *Klebsiella* resistance to polymyxins. To generalize that the presence of a hydroxyl group on a lipid A secondary acyl chain is a bacterial mechanism to evade innate immune defences warrants further studies. However, and providing additional support to this notion, hydroxylation on the 3'-linked secondary acyl chain of *Vibrio cholerae* also promotes resistance to antimicrobial peptides (Hankins et al Mol Microbiol. 2011 81(5): 1313–1329). Interestingly, the fact that other Gram negative pathogens also synthesize lipid A species that possess a hydroxyl group on a secondary acyl chain ([*Salmonella*, Gibbons et al J Biol Chem. 2000;275:32940–32949], [*Pseudomonas*, Kulshin et al Eur J Biochem. 1991;198:697–704] [*Legionella*, Zharinger et al Prog Clin Biol Res. 1995;392:113–139], [*Acinetobacter*, Beceiro et al Antimicrob Agents Chemother. 2011 55(7):3370-9]) could indicate that this lipid A modification is an evolutionary conserved mechanism playing a role in host survival; reinforcing the notion that this lipid A modification may play a more important role in polymyxins/antimicrobial peptide resistant than previously anticipated. In this context, we do agree with the referee's view that additional investigations testing different pathogens are required to explain mechanistically its role in resistance, and most likely also, in outer membrane stabilization. In the revised version of the discussion, we have included part of this discussion and expanded briefly what we hypothesize could be the role of this modification (lines 400-408).

and there is no hypothesis of how this new lipid A led can bind to TLR4/MD2 binary system in a different fashion, here maybe a deeper look at literature is worth. Maybe a couple of statements and a hypothesis could be advanced.

Following the referee's advice, we have included in the discussion (lines 459-470) new text summarizing very briefly how *mgrB*-controlled lipid A structure may limit TLR4/MD2 activation. Our speculations are based on the seminal work by Park and coworkers showing the structural basis of LPS recognition by TLR4/MD2 complex (Park et al *Nature* 458, 1191-1195).

Referee 2:

An exceptional quality study that establishes functional and mechanistic links between mutational modifications of lipid A to specific genes and lipid A structures in K. pneumoniae and in the model hosts. The only minor suggestion is to rephrase the statement regarding the "conventional wisdom"

or "dogma" that antibiotic resistance is linked to reduced bacterial fitness. This wisdom has been challenged in many publications in the last five years and by the simple fact of rapid spread in antibiotic resistance in clinics.

We thank the referee's comments on our work.

As suggested we have rephrased the statement concerning the "dogma" that antibiotic resistance is linked to reduce bacterial fitness. Additionally, and as indicated by the Editor, we have expanded the clinical implications of our work highlighting the notion that antibiotic resistance is not inexorably linked to reduce fitness but may result even in increase virulence (lines 480-496), as this manuscript demonstrates.

2nd Editorial Decision

09 January 2017

Thank you for the submission of your revised manuscript to EMBO Molecular Medicine. I am pleased to inform you that we will be able to accept your manuscript pending the following final amendments:

Ethics:

please indicate the age of the mice used and each instances, the exact n

Please submit your revised manuscript within two weeks. I look forward to seeing a revised form of your manuscript as soon as possible.

2nd Revision - authors' response

11 January 2017

Authors made requested editorial changes.

Corresponding Author Name: Professor Jose Bengoechea

Manuscript Number: EMM-2016-07336